# Benchmarking Spatiotemporal Reasoning in LLMs and Reasoning Models: Capabilities and Challenges

**Pengrui Quan, Brian Wang, Kang Yang, Liying Han, Mani Srivastava***
Department of Electrical and Computer Engineering, UCLA
{prquan, wangbri1, kyang73, liying98, mbs}@ucla.edu

## Abstract

Spatiotemporal reasoning plays a key role in Cyber-Physical Systems (CPS). Despite advances in Large Language Models (LLMs) and Large Reasoning Models (LRMs), their capacity to reason about complex spatiotemporal signals remains underexplored. This paper proposes a hierarchical *SpatioTemporal reAsoning benchmaRK*, *STARK*, to systematically evaluate LLMs across three levels of reasoning complexity: state estimation (e.g., predicting field variables, localizing and tracking events in space and time), spatiotemporal reasoning over states (e.g., inferring spatial and temporal relationships), and world-knowledge-aware reasoning that integrates contextual and domain knowledge (e.g., intent prediction, landmark-aware navigation). We curate 26 distinct spatiotemporal tasks with diverse sensor modalities, comprising 14,552 challenges where models answer directly or by Python Code Interpreter. Evaluating 3 LRMs and 8 LLMs, we find LLMs achieve limited success in tasks requiring geometric reasoning (e.g., multilateration or triangulation), particularly as complexity increases. Surprisingly, LRMs show robust performance across tasks with various levels of difficulty, often competing or surpassing traditional first-principle-based methods. Our results show that in reasoning tasks requiring world knowledge, the performance gap between LLMs and LRMs narrows, with some LLMs even surpassing LRMs. However, the LRM o3 model continues to achieve leading performance across all evaluated tasks, a result attributed primarily to the larger size of the reasoning models. STARK motivates future innovations in model architectures and reasoning paradigms for intelligent CPS by providing a structured framework to identify limitations in the spatiotemporal reasoning of LLMs and LRMs.

## 1  Introduction

Cyber-Physical Systems (CPS) play a critical role in numerous daily and industrial applications, enabling applications such as precision agriculture, environmental monitoring, and smart city infrastructure. Effective operation in these domains relies on spatiotemporal intelligence: the ability to perceive dynamic environments, understand spatiotemporal relationships, and integrate this understanding with world knowledge to make informed decisions. For instance, mobile robots need to fuse sensor data (e.g., range sensor) to navigate cluttered spaces. For various CPS applications, the common denominator is the necessity for reasoning jointly over space, time, and world knowledge.

Recent progress in large language models (LLMs) and large reasoning models (LRMs) [19, 11, 8] suggests their potential as general-purpose agents for CPS applications. Yet the community lacks rigorous benchmarks that consider the complexity in CPS applications. As shown in Figure 1,

---

*Mani Srivastava holds concurrent appointments as a Professor of ECE and CS (joint) at the University of California, Los Angeles, and as an Amazon Scholar at Amazon. This paper describes work performed at UCLA and is not associated with Amazon.

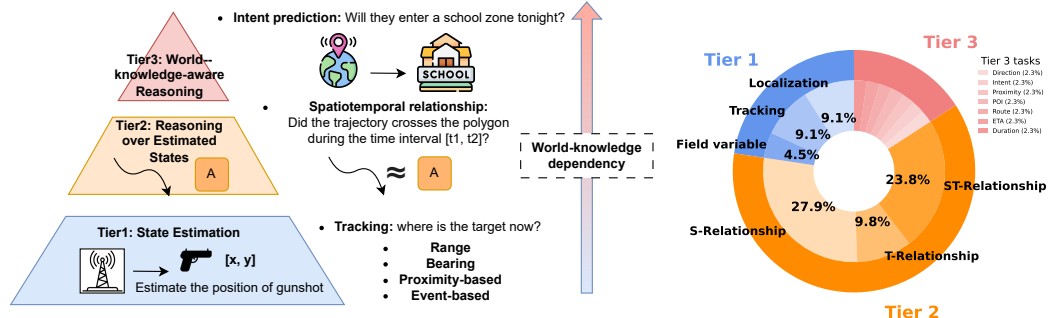

Figure 1: Three-Tiered Architecture of STARK.

spatiotemporal reasoning in CPS typically proceeds through three stages: (1) First, state estimation. The system must perceive its environment and understand the current situation. This step involves determining what is happening, where, and when. (2) Reasoning over estimated states. Once a set of current and past states is estimated, the system needs to understand how these states interrelate across space and time. (3) Finally, world-knowledge-aware reasoning. For complex real-world scenarios, the estimated states and their spatiotemporal relationships are often interpreted with contextual information and world knowledge to make more nuanced inferences, predictions, or decisions.

However, existing benchmarks [16, 29, 17] fall short of capturing this comprehensive, structured view of spatiotemporal reasoning. To address this gap, we introduce STARK (SpatioTemporal reAsoning benchmaRK), a hierarchical benchmark that measures how well models can plan and execute spatiotemporal reasoning pipelines. Besides, in addition to direct answering, our benchmark is specifically designed to assess tool usage competency, viewing it as an essential move to benchmark real-world CPS agents. We designed STARK with the following features:

- **Task novelty.** STARK includes under-explored problems such as field-variable prediction, spatiotemporal localization, and tracking that intertwine spatial geometry and temporal operation.

- **Task rigor.** STARK requires models to craft and explain structured solution strategies. While established methods exist (e.g., `multilateration`), the goal of STARK is to assess whether models can identify and correctly apply the appropriate techniques for spatiotemporal tasks.

- **Multi-level evaluation.** STARK is structured across three levels: (i) *state estimation*, (ii) *reasoning over estimated states*, and (iii) *world-knowledge-aware reasoning* that incorporates both algorithmic computation and external knowledge. This separation enables a fine-grained assessment of model capabilities across increasing levels of complexity and world-knowledge dependency.

- **Task difficulty.** Unlike prior multiple-choice suites [16], STARK includes open-ended challenges: the model must output a numeric estimate given the reasoning challenges instead of picking from canned answers. Specifically, we evaluate the case where the model **directly answers (DA)** or invokes a Python **code interpreter (CI)** supplied at inference time. The CI mode serves as a primary metric for evaluating the model's capability to use external, deterministic tools, a core requirement for reliable CPS agents. This design effectively exposes both the model's spatiotemporal reasoning ability and its competency in tool usage.

Overall, our study makes the following three primary contributions:

- Comprehensive evaluation. We release STARK with 26 spatiotemporal reasoning scenarios and 14,552 challenge instances, providing a broad picture of LLM and LRM performance on spatiotemporal reasoning (Figure. 4). Our evaluation shows promising abilities of LRMs while also revealing their current limitations.

- Analysis of models. We benchmark 8 LLMs and 3 LRMs, explore how **direct answering (DA)** and **code interpreter (CI)** modes affect performance, and identify limitations in spatiotemporal reasoning. Besides, we compare the models with existing first-principle baselines.

- Open resources. The benchmarks, STARK-L (14k samples for comprehensive evaluation [2]) and STARK-S (1.3k samples for rapid usage[3]), and code[4] are publicly available to facilitate reproducibility and foster more capable spatiotemporal reasoning systems. By exposing where LLMs and LRMs excel and fall short, STARK can inspire more effective solutions for intelligent CPS applications.

## 2 Related works

| Benchmark | Sensor type | Spatial | Temporal | Field analysis | Localization | Tracking | Open-end | Mode |
|---|---|---|---|---|---|---|---|---|
| GeoQA[28] | Coordinate | ✓ | ✗ | ✗ | ✓ | ✗ | ✗ | DA |
| GeoQA[17] | Coordinate | ✓ | ✗ | ✗ | ✗ | ✗ | ✗ | DA |
| TemporalQA[24] | Text | ✗ | ✓ | ✗ | ✗ | ✗ | ✗ | DA |
| Open3DVQA[29] | Image | ✓ | ✗ | ✗ | ✓ | ✗ | ✗ | DA |
| STBench[16] | Coordinate | ✓ | ✓ | ✗ | ✗ | ✗ | ✗ | DA |
| STARK | Diverse | ✓ | ✓ | ✓ | ✓ | ✓ | ✓ | DA & CI |

Table 1: Overview of existing benchmarks (DA: directly answers; CI: code interpreter)

**Spatial, temporal, and spatiotemporal benchmark.** GeoQA [28, 17] evaluates symbolic (e.g., north-of) and geometry reasoning but ignores interaction with different sensor modalities and temporal reasoning. Temporal-QA [24, 26] focus on event ordering and interval logic in pure text, yet they do not include spatial constraints or open-ended numeric estimation. STBench[16] combines both spatial and temporal axes but either collapses tasks into multiple-choice or ignores other common spatial estimation tasks, such as field variable prediction, localization, or tracking. In addition, it lacks the evaluation of a spatiotemporal logic framework (Tier 2 in STARK) and reasoning with both algorithmic computation and world knowledge (Tier 3 in STARK). Vision-LLMs [3, 29] are evaluated on spatiotemporal reasoning tasks using egocentric videos or images, focusing on object recognition, temporal event ordering, and coarse spatial grounding. However, it focuses on a single deployed sensor and lacks explicit geometric computation and multi-sensor reasoning. Lastly, none of the above works attempt to evaluate LRMs or the effectiveness of using CI.

**Spatial and temporal logic frameworks.** Spatial and temporal relations in STARK build directly on two classical frameworks that formally describe spatial and temporal operations. For space, we adopt the dimensionally-extended nine-intersection model (DE-9IM) [4, 5] implemented in ArcGIS [21, 12], whose topological predicates exhaustively categorize how any two geometries relate and form the backbone of modern GIS query engines, including `intersects`, `contains`, `within`, `touches`, `overlaps`, and their complements. For time, we rely on Allen's interval algebra [2], whose 13 atomic relations provide a mutually exclusive, jointly exhaustive vocabulary for reasoning over intervals on a timeline (`before`, `meets`, `overlaps`, `during`, `starts`, `finishes`, `equals` and their converses). Together these frameworks constitute the symbolic operations for spatiotemporal inference. To our knowledge, no prior benchmark has directly evaluated whether LLMs can reason over both ArcGIS spatial predicates and Allen's interval algebra in a structured setting. STARK fills this gap by incorporating both formalisms into open-ended, sensor-grounded tasks where models must apply formal spatial and temporal logic to realistic trajectories and interval structures.

## 3 STARK

In contrast to existing benchmarks that offer fragmented views of spatiotemporal reasoning, this section introduces STARK. Table 2 summarizes tasks, sensor modalities, and example questions.

### 3.1 Sensor modalities

To support realistic and diverse spatiotemporal tasks in STARK, we simulate five representative sensor modalities in Figure 2: **Range** (distance measurement, enabling `multilateration`), **Bearing** (angular measurement, supporting `triangulation`), **Range & Bearing** (jointly output both distance and angle), **Proximity** (binary output for coarse localization via set intersection), and **Event-based**

---

[2]`https://huggingface.co/datasets/prquan/STARK_10k`
[3]`https://huggingface.co/datasets/prquan/STARK_1k`
[4]`https://github.com/nesl/STARK_Benchmark/`

| Tier | Task | Sensor modality | Examples |
|---|---|---|---|
| 1 | Spatiotemporal forecast
Spatiotemporal impute
Temporal impute
Spatial impute | Air quality, traffic,
& temperature | What is the value of sensor reading at time t? (18) |
| | Spatial Localization
Spatial Localization
Spatial Localization
Spatial Localization
Spatial Localization | Range
Bearing
Range & bearing
Region
Event | Where is the object most likely located? Provide the estimated [x, y] coordinates. (14) |
| | Temporal Localization | Event | Estimate when the seismic event occurred. (15) |
| | Spatial tracking
Spatial tracking
Spatial tracking
Spatial tracking
Spatial tracking | Range
Bearing
Range & bearing
Region
Event | Based on the available measurements at each time stamp (including the historical data), estimate the most likely [x, y] coordinates of the object in the 2D plane. (16) |
| | Temporal tracking | Event | Based on the given TOA data, estimate when the shot was fired at each step. (17) |
| 2 | Spatial relationship | Coordinate | Determine whether the {geometric object 1} has the spatial relationship **{relate}** with the {geometric object 2} (19) |
| | Temporal relationship | Coordinate | Determine whether the time interval {t_1} has the temporal relationship **{relate}** with the time interval {t_2} (20) |
| | Spatiotemporal relationship | Coordinate | Determine whether the time interval during which the EVENT holds has the temporal relationship **{temporal_relationship}** with the reference interval {interval_2}. EVENT: {event_1} (21) |
| 3 | Landmark direction | Text + Coordinate | Determine whether the most accurate spatial relationship between {landmark_1} and {landmark_2} is {direction}, selecting from the options: ['north of', 'north-east of', 'east of', 'south-east of', 'south of', 'south-west of', 'west of', 'north-west of'] (22) |
| | Landmark proximity | Text + Coordinate | Is there a {amenity} within {distance} metres of {landmark}? (23) |
| | Route planning | Text + Coordinate | Does the route from {landmark_1} to {landmark_2} pass by {landmark_3}? Route: {route} (24) |
| | ETA calculation | Text + Coordinate | If I leave {landmark_1} by car at {t1}, can I arrive at {landmark_2} by {t2}? (25) |
| | Route segment duration | Text + Coordinate | I drove from {landmark_1} at {t1}, and the trip took about {D} minutes. Was I likely to witness the accident that occurred along the route from {p_1} to {p_2} between {t2} and {t3}? Answer directly based on the reasoning of spatial and/or temporal information. (26) |
| | POI prediction | Text + Coordinate | Determine if the user is likely to visit location {x} in the next 5 hrs? (27) |
| | Intent prediction | Text + Coordinate | Determine if the user is likely to {do_X} in the next 5 hrs? (28) |

Table 2: Overview of the benchmark organization

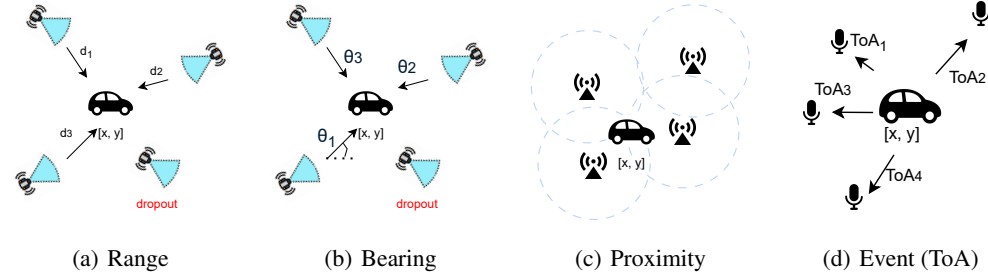

(a) Range      (b) Bearing      (c) Proximity      (d) Event (ToA)

Figure 2: State estimation. Four types of sensor modalities for localization and tracking.

(timestamped detections for spatiotemporal inference using time-of-arrival differences). These modalities reflect common real-world sensing systems and elicit a range of geometric and combinatorial reasoning strategies. Sensor readings are injected with modality-specific noise (e.g., 1% for range sensors to mimic LiDAR accuracy; full specifications in Appendix A.4).

## 3.2 Task generation

### 3.2.1 State estimation

**Field variable prediction** tasks aim to estimate sensor measurements across space and time, including spatial imputation (predicting a sensor's value from neighboring sensors at the same time), temporal imputation (using historical and future values), spatiotemporal imputation (leveraging neighboring sensors' histories), and spatiotemporal forecasting (predicting future values for neighboring sensors). We use real-world air quality data from PurpleAir [1] and traffic data from CalTrans PeMS [23], and also synthesize temperature readings with controlled spatial properties.

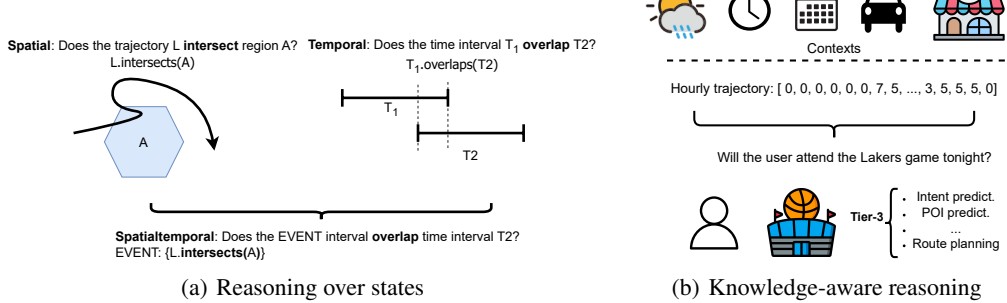

(a) Reasoning over states      (b) Knowledge-aware reasoning

Figure 3: Illustration of Tier 2 and Tier 3 challenges

**Spatial localization** aims to estimate an object's location from sensor measurements. Each task is generated by randomly placing four sensors and a target within a $10 \times 10$ grid, with all sensors sharing the same modality. Measurements are simulated (with noise) based on the target's true position and provided to the model, which predicts the target's $[x, y]$ location.

**Temporal localization** estimates the time of an event from detection times recorded by multiple spatially distributed event-based sensors. Using the same setup as spatial localization, the model's task is to infer the event's actual time of occurrence from these noisy arrival times.

**Spatial tracking** addresses dynamic object localization along continuous trajectories. In our established four-sensor, $10 \times 10$ plane simulation, the object movement produces a time-stamped trajectory and sensor measurement. Inspired by SCAAT tracking [27], we introduce partial observability via random sensor dropout. This forces the model to handle incomplete, noisy data (e.g., only two active sensors) by historical trajectory patterns and motion continuity to infer position, reflecting real-world challenges in autonomous navigation. Appendix A.3 details trajectory simulation.

**Temporal tracking** estimates the timing of events occurring along a moving trajectory. We adopt the same prior sensor and environment configuration similar to SCAAT [27].

### 3.2.2 Reasoning over states

**Spatial relationship reasoning** evaluates a model's ability to infer qualitative spatial relationships between geometric objects. Spatial predicates defined in ArcGIS [12, 21] lead to 35 possible combinations of geometry pairs and spatial relations (Appendix A.5). For each instance, a pair of geometric objects is sampled and a specific relationship is generated using Shapely[7].

**Temporal interval reasoning** tests a model's ability with Allen's interval algebra [2]. We generate tasks by simulating two random intervals, labeling their temporal relationship, and creating a balanced negative counterpart.

**Spatial-temporal reasoning** evaluates a model's ability to combine spatial and temporal logic. Using trajectories from spatial tracking, we identify intervals where a spatial predicate (e.g., Intersects, Within) holds between a trajectory and a geometry object. These event intervals are paired with reference intervals, and the model needs to decide whether a specified temporal relation (e.g., Overlaps, Meets) holds, as shown in Figure 3 (a). This task requires the model to localize events in time via spatial constraints and reason about their temporal relationships.

### 3.2.3 Knowledge-aware reasoning

Using ArcGIS [12, 21] and OpenStreetMap [9], we produce the following challenges that require world knowledge consideration: (1) **Landmark direction** evaluates a model's ability to infer directional relationships between real-world landmarks. (2) **Landmark proximity** tests a model's spatial reasoning in urban contexts by reasoning the presence of amenities near landmarks (e.g., Is there a police station within 500 meters of Central Park?). (3) **Route planning** tests whether models can connect textual travel descriptions to real-world spatial layouts. The model must decide if a given route passes by a specified landmark. (4) For **ETA calculation**, the model is asked to determine, given a departure time and average speed, if arrival before a specified deadline is possible based

only on textual descriptions. (5) **Route segment duration** test the model on whether a user could plausibly be present along the segment during a given time window.

Additionally, we resort to simulating user trajectories for controlled ground-truth user intent to address data scarcity and privacy concerns in real-world mobility datasets. (6) **POI prediction** simulates human mobility using a finite state machine (FSM)[6, 14], where each state represents a point of interest (POI). We generate a synthetic seven-day trajectory at hourly intervals. To enhance realism, we integrate contextual factors, such as weather, time, traffic, social events, etc., which are also expressed in natural language, into the models. The model must predict the next likely POI, requiring reasoning over **spatiotemporal traces** and **language context** (Figure 3 (b)). (7) Similarly, **intent prediction** extends POI prediction by requiring the model to infer the user's underlying intent behind each visit. The model must predicts activity category (e.g., going home, watching a movie).

### 3.3 Baseline generation

We establish first-principle baselines for Tier 1 tasks using theoretically grounded methods for localization, tracking, and field variable prediction (details in Appendix A.6). For Tier 2 and Tier 3 tasks, however, many problems are solvable deterministically or through oracle-level services. For instance, spatial reasoning tasks such as point-in-polygon, intersection, or containment can be exactly solved using spatial libraries like Shapely [7], while world-knowledge-based queries (e.g., nearest landmark, directional relation) can be precisely answered by GIS services such as ArcGIS [12]. As these tools achieve near-perfect accuracy, including them as baselines would render comparisons with LLMs/LRMs uninformative. Instead, we report a random-guess baseline (0.5 accuracy under balanced labels) to reflect the challenge of learning such reasoning capabilities from language and context without deterministic priors.

## 4 Experiments

### 4.1 Experiment setup

We conduct evaluations on both versions of the STARK benchmark. Specifically, we evaluated STARK-S (partial, 1.3k samples) and STARK-L (full, 14k samples), and we report results on STARK-S in the main paper and refer readers to the Appendix (Table 10 and 8) for detailed results on STARK-L and STARK-S. All system prompts are provided in Table 3. Cost information is shown in Table 4 to support dataset selection.

For all tasks except field variable prediction, we report the Root Mean Squared Error (RMSE) against the ground truth (for both open-ended calculation and true or false questions). For field variable prediction, where outputs vary in scale, we instead report the Root Mean Squared Percentage Error (RMSPE), because of differences in data magnitude across datasets. To ensure ro-

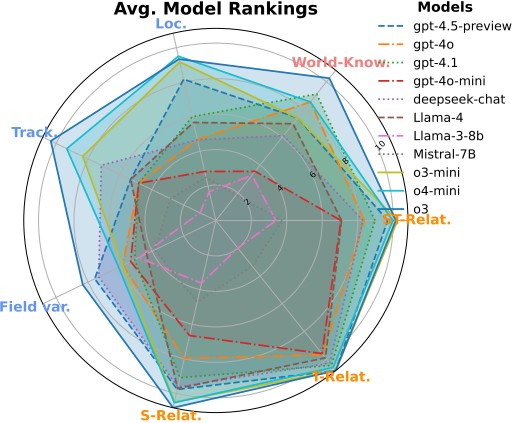

Figure 4: Model ranks. Further from the center indicates better performance.

bustness to extreme outliers, we compute all metrics using the trimmed values unless stated otherwise, i.e., excluding the top and bottom 10% of the distribution.

For evaluating the models using a code interpreter (CI), we execute the code in a sandbox environment and report the relative change in error compared to direct answering (DA). This is calculated as the percentage change in RMSE and RMSPE due to using CI (lower values indicate better performance). Code execution was supported using `scipy`, `numpy`, `shapely`, and `filterpy` libraries [25, 10, 7, 15].

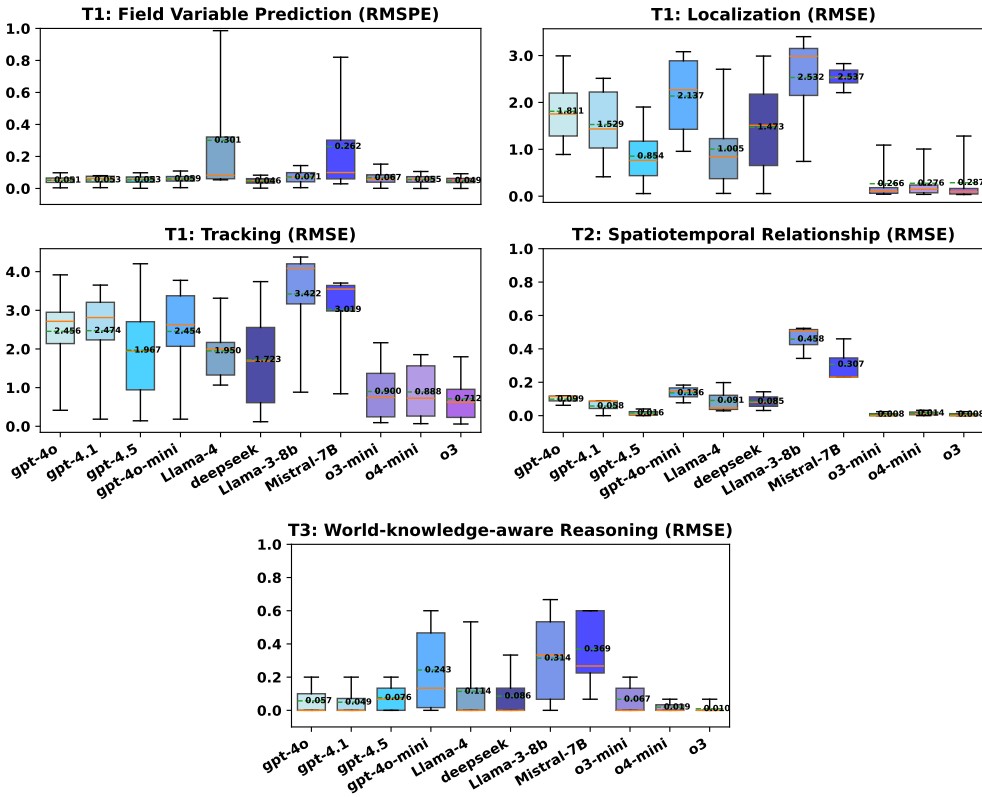

Figure 5: Benchmark results aggregated over task. Model performance is measured by RMSPE (field variable prediction) and RMSE (others). The numbers in each box plot represent the median.

## 4.2 Model evaluation

Figure 5 and 4 present the quantitative comparison between LRMs (o3, o3-mini, o4-mini) and LLMs (others). We refer readers to Table 8 in the appendix for the exact task-wise errors. We observe that LRMs consistently outperform LLMs on Tier 1 and Tier 2 tasks, which include localization, tracking, and spatiotemporal relationship reasoning. The performance gap is especially pronounced in localization and tracking tasks, where o3 achieves $3\times$ to $10\times$ lower error than LLM counterparts. Among LLMs, GPT-4.5 demonstrates the strongest overall reasoning capabilities in localization and spatial/temporal relationship inference, while deepseek-chat excels in field variable prediction and tracking. Despite their strong performance in spatiotemporal reasoning, smaller LRMs such as o3-mini lag behind LLMs on Tier 3 tasks involving world knowledge, such as intent prediction and POI classification. Notably, the full o3 model maintains strong performance even in these knowledge-intensive tasks, highlighting the importance of model scale for LRMs in applying world knowledge. Since the model sizes are not disclosed, we speculate that o3 is approximately $10\times$ larger than the o3-mini family, given the cost is also about $10\times$ higher [18].

## 4.3 First-principle baselines

**Localization.** Figure 6 compares optimization-based localization approaches vs. LLMs across six tasks. Firstly, optimization-based methods generally outperform LLMs, particularly for range-only and bearing-only tasks, due to their reliance on precise geometric optimization algorithms. However, in certain instances, LRMs can match or surpass baselines. For example, in the localization range task, o3 performs the best. The baseline falters, largely due to multilateration's sensitivity to sensor placement and non-convex optimization (Figure 7), which is exacerbated by Geometric Dilution of Precision (GDOP) when sensors are poorly arranged.

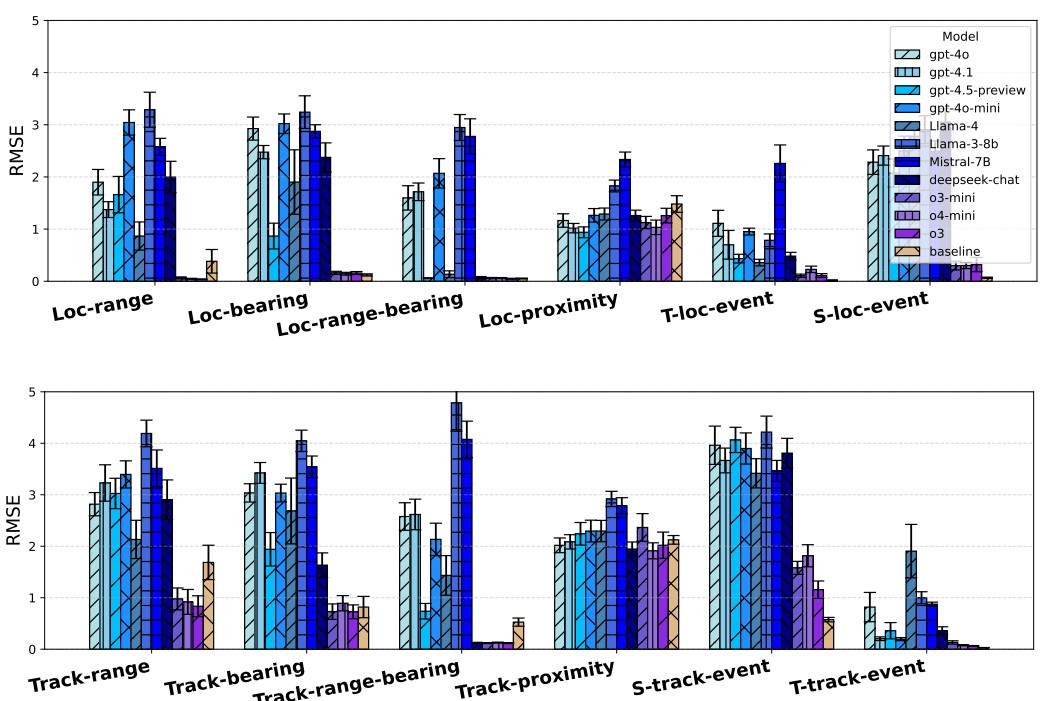

Figure 6: Localization (top) and tracking (bottom) compared with baselines.

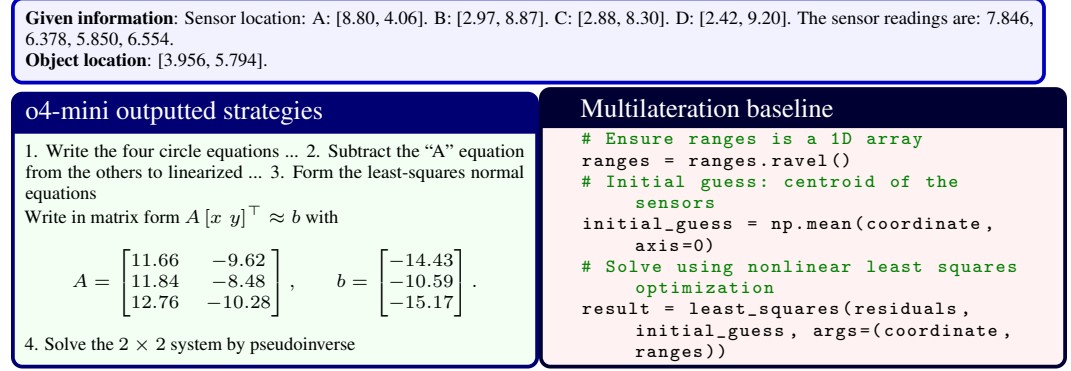

**Given information**: Sensor location: A: [8.80, 4.06]. B: [2.97, 8.87]. C: [2.88, 8.30]. D: [2.42, 9.20]. The sensor readings are: 7.846, 6.378, 5.850, 6.554.
**Object location**: [3.956, 5.794].

### o4-mini outputted strategies

1. Write the four circle equations ... 2. Subtract the "A" equation from the others to linearized ... 3. Form the least-squares normal equations

Write in matrix form $A\,[x\ y]^\top \approx b$ with

$$A = \begin{bmatrix} 11.66 & -9.62 \\ 11.84 & -8.48 \\ 12.76 & -10.28 \end{bmatrix}, \qquad b = \begin{bmatrix} -14.43 \\ -10.59 \\ -15.17 \end{bmatrix}.$$

4. Solve the $2 \times 2$ system by pseudoinverse

### Multilateration baseline

```
# Ensure ranges is a 1D array
ranges = ranges.ravel()
# Initial guess: centroid of the
    sensors
initial_guess = np.mean(coordinate,
    axis=0)
# Solve using nonlinear least squares
    optimization
result = least_squares(residuals,
    initial_guess, args=(coordinate,
    ranges))
```

Figure 7: Range-based localization examples. The four sensors do not surround the target, and three are clustered together. This configuration leads to a poorly conditioned Jacobian matrix for the range equations, meaning that small errors in the sensor measurements can cause large uncertainties. Because of this, an optimization-based technique (right) may struggle to converge to a good minimum. Instead, the model chooses a Pseudo-inverse solution, aiming to mitigate the GDOP impact.

**Tracking.** Figure 6 shows the performance of LLMs and Extended Kalman Filter (EKF) baselines [13] across various sensor modalities. Overall, classical EKF-based baselines outperform most LLMs by leveraging recursive filtering and explicit motion models. However, in the range and bearing tracking task, LRMs often surpass the baseline. This is mainly due to (i) the high sensitivity of bearing measurements to noise, (ii) random sensor dropout resulting in large uncertainty, and (iii) the baseline's sensitivity to poor initialization of object location, which can cause rapid error accumulation.

As shown in Figure 8, LRMs' consistent robustness likely stems from their ability to integrate partial observations, learn trajectory patterns, and apply contextual reasoning, maintaining accurate tracking even with ambiguous or missing data. Lastly, we observe similar results in field variable prediction tasks (Figure 11).

**Given information**:
Sensor location: A: [0.35, 5.77]. B: [6.58, 8.34]. C: [5.92, 4.49]. D: [8.29, 9.01].
Step 0 | Time 0.000, sensor readings: [nan, nan, 53.6111, nan].
Step 1 | Time 0.2041, sensor readings: [nan, nan, nan, 27.3684].
**Step 2 | Time 0.4082, sensor readings: [19.1300, 25.6084, nan, 25.5940].**
Step 3 | Time: 0.612, sensor readings: [nan, nan, nan, 15.8711].
**Step 4 | Time: 0.816, sensor readings: [nan, 19.5208, 56.7513, 12.8046].**
...
**Object location**:
Step 0 | [10.0, 10.0]. Step 1 | [9.796 9.796]. **Step 2 | [9.592 9.592].** Step 3 | [9.388 9.388]. **Step 4 | [9.184 9.184]**...

| Outputs of o4-mini | Extended Kalman Filter (EKF) baseline |
|---|---|
| Step 0 \| LRM: [6.51, 5.30] | Step 0 \| EKF: [7.621 7.517] |
| Step 1 \| LRM: [5.37, 7.50] | Step 1 \| EKF: [11.040 6.512] |
| **Step 2 \| LRM: [4.02, 7.04]** | Step 2 \| EKF: [13.458 11.485] |
| Step 3 \| LRM: [2.45, 7.33] | Step 3 \| EKF: [16.621 12.051] |
| **Step 4 \| LRM: [8.94, 9.17]** | Step 4 \| EKF: [19.035 13.431] |
| ... | ... |

Figure 8: Bearing-based tracking examples. When the initial object location is unknown and only one sensor is available at the first two steps, the LRM makes an inaccurate estimation. However, in contrast to EKF, LRM discards the inaccurate prior measurements and starts performing triangulation at the third and fifth steps.

**Given information**:
Sensor location: - Sensor A: [9.35, 9.99]. Sensor B: [6.75, 2.85]. Sensor C: [5.69, 5.45]. Sensor D: [5.94, 9.16]
Step 0 | The corresponding sensor readings are 0.3394, 0.1314, 0.0379, 0.2101.
Step 1 | The corresponding sensor readings are 0.6963, 0.7627, 0.6333, 0.5806.
**Step 2 | The corresponding sensor readings are 1.0990, 1.3683, 1.2629, 1.1270.**
...
**Object location**:
Step 0 | [5.00 5.00]. Step 1 | [6.49 7.85]. **Step 2 | [7.85 9.69]**...

| Outputs of o4-mini written program | Model written code |
|---|---|
| Step 0 \| CI: [4.82, 4.88] | |
| Step 1 \| CI: [6.33, 7.83] | |
| **Step 2 \| CI: [-12246, 32902]** | |
| Step 3 \| CI: [2.45, 7.33] | |
| ... | |
| **RMSE: 7847** | |

```python
def residuals(x):
    shooter = x[0:2]
    T0 = x[2]
    dists = np.linalg.norm(sensors_used - shooter,
        axis=1)
    t_pred = dists/v + T0
    return t_pred - t_meas
# Use the sensor with the earliest arrival time for
    an initial guess.
idx_min = np.argmin(t_meas)
```

Figure 9: CI failure example. The code written by the model is sensitive to initialization, suggesting the need for a feedback mechanism to avoid invalid answers and improve solution quality.

### 4.4 Result of using a code interpreter (CI)

Figure 10 summarizes the relative change in RMSE when models answer questions through Python coding with libraries compared to directly answering (DA). To mitigate extreme outliers resulting from occasional large errors produced by Python code execution, we calculated the trimmed RMSE/RMSPE for both DA and CI, removing the top and bottom 25% of data points. Several key observations emerge: (1) For localization tasks, CI notably enhances the performance of LLMs such as gpt-4o, gpt-4.1, and gpt-4o-mini, achieving error reductions of nearly 80%. (2) Conversely, for tracking tasks, CI generally degrades performance. This degradation occurs because LLM-generated tracking algorithms are vulnerable to suboptimal sensor placement, resulting in significant errors that are less frequent in DA scenarios. (3) For the LRM o3-mini, the benefits from CI are marginal overall, though occasionally coding leads to substantial increases in error (up to +86%) due to optimization-related inaccuracies. To mitigate such severe errors as shown in Figure 9, it may be necessary to implement additional mechanisms, such as guardrails with rule-based feedback and iterative refinement, to rule out the severe errors.

## 5 Conclusion and future work

We present a comprehensive benchmark of 26 spatiotemporal reasoning tasks to evaluate 8 LLMs and 3 LRMs. Our findings indicate that (1) LLMs could struggle with complex geometric and sensor-driven tasks, whereas LRMs demonstrate robust performance, frequently outperforming

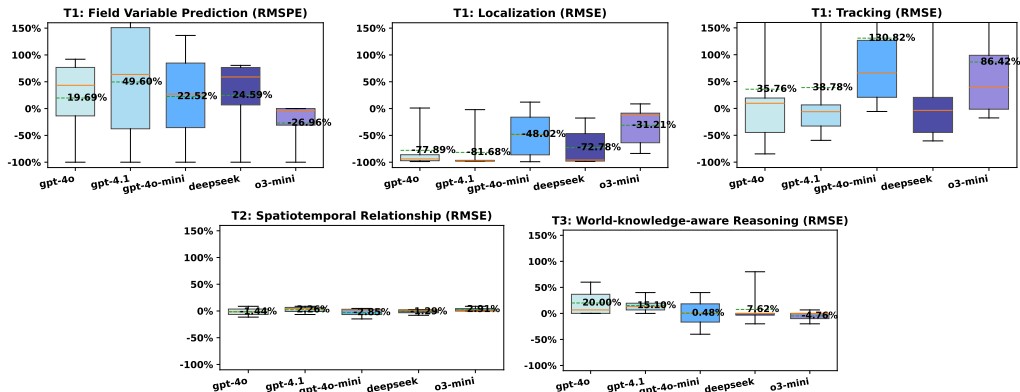

Figure 10: Results of using CI. The table shows the relative change in RMSPE/RMSE compared with DA. The absolute performance is reported in Table 9. The numbers in each box plot represent the median.

classical first-principle baselines in localization and tracking. (2) We further explored the impact of model size on reasoning capability and observed a clear correlation. Larger LRM o3 exhibited superior performance across all tasks. Conversely, smaller LRMs such as o3-mini and o4-mini displayed limitations in integrating world knowledge, highlighting the role of model capacity in complex reasoning tasks. (3) We evaluated the impact of Code Interpreter (CI) on model performance. Interestingly, coding capabilities could enhance and diminish LLM effectiveness, suggesting the need for well-defined guardrails or rule-based constraints to guide model outputs.

Several promising avenues remain for future research. As shown in our Code Interpreter (CI) analysis, open-ended code execution can both enhance and destabilize performance. To address this, future work could develop guardrails and debugging layers for CI-based reasoning (detecting self-consistency or retrying with a feedback loop). Additionally, considering the demonstrated success of LRMs, applying reinforcement learning [8, 22] to fine-tune LLMs specifically for spatiotemporal reasoning tasks represents a compelling research direction. Such fine-tuning could substantially enhance the reasoning capabilities of LLMs, potentially narrowing the performance gap between general-purpose LLMs and specialized LRMs. These future directions could motivate advancements in model architectures and reasoning paradigms for intelligent CPS.

## 6    Acknowledgment

This research was sponsored in part by AFOSR award #FA95502210193, DEVCOM ARL award #W911NF1720196, NSF award #CNS-2325956, NIH award #P41EB028242, and Sandia National Laboratories award #2169310. Mani Srivastava was also partially supported by the Mukund Padman-abhan Term Chair at UCLA.

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

# A  Appendix

## A.1  System prompts

| Mode | System prompt |
|---|---|
| DA | You are a helpful AI assistant. No coding or tools are allowed when you help users. |
| CI | You are a helpful AI assistant. |
| | Instructions: |
| | 1. Python Coding: Use Python codinng for signal processing tasks. Implement your functions inside "'Python "' code block. Do not write code outside the functions. The function prototypes are as follows: |
| | You need to implement the function the solver (mandatory): |
| | "'Python |
| | # If you use the print() function, the output will be brought to you. # You're running python in a non-interactive environment, the variable name alone will not output anything. # You need to implement the function solver. You can only use python libraries numpy, shapely, and scipy. |
| | def solver(): |
| | # HERE is where you put your solution. # import necessary libraries such as numpy or scipy here # input: None. # output: result: an numpy array storing the result |
| | pass |
| | return result |
| | "' |
| | 2. [IMPORTANT] State your answer between keywords [RESULTS_START] and [RESULTS_END], and the iteration will stop. Output [RESULTS_START] and [RESULTS_END] in the chat directly. |

Table 3: System prompts for DA and CI

## A.2  Benchmarking cost

| Model | STARK-S | STARK-L | Provider |
|---|---|---|---|
| O3 | $418.5 | $759.8[*] | OpenAI |
| O3-mini | $36.2 | $442.9 | OpenAI |
| O4-mini | $23.0 | $352.6 | OpenAI |
| GPT-4.5 | $320.6 | -[†] | OpenAI |
| GPT-4.1 | $21.4 | $237.5 | OpenAI |
| GPT-4o | $32.9 | $284.1 | OpenAI |
| LLaMA-4 | $7.39 | $44.4 | Together.ai |
| LLaMA-3-8B | $0.27 | $9.21 | Together.ai |
| Mistral-7B | $0.69 | $25.1 | Together.ai |

[*]Due to cost reduction by OpenAI, the benchmarking cost of o3 has been reduced by 80% for STARK-L.
[†]We excluded GPT-4.5 (which is no longer accessible) since it has been deprecated by OpenAI.

Table 4: Cost of benchmarking each model once with DA.

## A.3  Object trajectory simulation

To simulate diverse object trajectories in a two-dimensional plane, we construct synthetic motion paths by linearly combining up to three randomly selected basis motions from a library of parametric models: linear, circular, sinusoidal, figure-eight, and spiral motions. Each motion type is defined by a set of physical parameters, such as initial position, velocity, radius, amplitude, angular velocity, and phase, which are sampled uniformly within specified ranges to ensure variety and coverage across the 2D domain. For each trajectory, a random subset of one to three motion types is chosen, and their respective parameter sets are generated independently. The resulting trajectories are computed by summing the contributions of the selected motions over a fixed time interval, sampled at a given rate. To ensure all simulated paths are constrained within the spatial domain $[0, 10] \times [0, 10]$, we apply min-max normalization to the x and y coordinates based on the computed extrema. The final output consists of a time series of normalized positions with diverse motion patterns. This framework enables controlled, reproducible generation of complex object trajectories.

## A.4 Noise model (Table 5)

| Simulation Scenario | Measurement Type | Parameters | Noise Injection Method |
|---|---|---|---|
| Spatial localization/tracking | Range | 0.01 | Uniform noise relative to true distance: $range\_measured = range_{true} + U(-range_{true} \cdot \sigma, range_{true} \cdot \sigma)$ |
| Bearing localization/tracking | Bearing | $\sigma=0.01$ | Uniform noise proportional to full circle (360°): $bearing\_measured = bearing_{true} + U(-2\pi\sigma, 2\pi\sigma)$ |
| Range-Bearing localization/tracking | Range | $\sigma=0.01$ | Uniform noise relative to true distance. Same as previous. |
| Region-Based localization/tracking | Region | $r = 4.0$ | No explicit numeric noise; discrete hit/miss criteria based on a sensor coverage radius $r$ |
| Temporal localization/tracking | Event/ToA | $\sigma = 0.01$ | Uniform noise independent of distance (absolute value) $time\_measured = time_{true} + U(-\sigma, \sigma)$ |
| Spatial localization/tracking | Event/ToA | $\sigma = 0.01$ | Uniform noise independent of distance (absolute value): $time\_measured = time_{true} + U(-\sigma, \sigma)$ |

Table 5: Summary of noise injection methods for each simulation scenario

## A.5 Simulating spatial relationship

| | Equals | Intersects | Contains | Within | Crosses | Touches | Overlaps |
|---|---|---|---|---|---|---|---|
| Point and point | ✓ | × | × | × | × | × | × |
| Point and linestring | × | ✓ | × | ✓ | × | ✓ | × |
| Point and polygon | × | ✓ | × | ✓ | × | ✓ | × |
| Linestring and point | × | ✓ | ✓ | × | × | ✓ | × |
| Linestring and linestring | ✓ | ✓ | ✓ | ✓ | ✓ | ✓ | ✓ |
| Linestring and polygon | × | ✓ | × | ✓ | ✓ | ✓ | × |
| Polygon and point | × | ✓ | ✓ | × | × | ✓ | × |
| Polygon and linestring | × | ✓ | ✓ | × | ✓ | ✓ | × |
| Polygon and polygon | ✓ | ✓ | ✓ | ✓ | × | ✓ | ✓ |

Table 6: Spatial relationship between geometries

## A.6 Baseline generation

We derive reliable reference points as baselines grounded in first principles and theoretically established solutions.

**Localization.** For range-only sensors, we employ a `multilateration` method that estimates the object's position by minimizing the residual between measured and predicted distances using nonlinear least squares. The bearing-only baseline relies on `triangulation` by formulating a linear system based on angle measurements and solving for the intersection point of bearing lines from multiple sensors. For proximity-based localization, we define a cost function that penalizes inconsistencies between observed binary detection (inside/outside a region) and estimated positions, then perform a grid search over the spatial domain to find the position minimizing this cost. The event-based localizations follow a similar optimization approach by minimizing the squared residuals between observed and predicted time-of-arrival values, assuming a known wave propagation speed.

**Field variable prediction.** For temporal imputation, we apply linear interpolation across the time dimension, while spatiotemporal forecasting is handled via straightforward extrapolation from recent observations. When imputing missing values at unseen spatial locations in spatiotemporal imputation, we adopt a distance-weighted averaging scheme, where nearby known values contribute proportionally based on their proximity to the target location.

**Tracking.** To establish tracking baselines for each sensor modality, we develop lightweight model-based algorithms grounded in classical filtering and geometric optimization techniques. For range-only tracking, we implement an SCAAT-style Extended Kalman Filter (EKF) [13, 20], where the object's states, position, and velocity are updated sequentially based on individual range measure-

| Task family | Scenario | Baseline approach (key idea) |
|---|---|---|
| Localization | Range-only | `Multilateration`: minimize nonlinear least-squares residual between measured and predicted ranges. |
| | Bearing-only | `Triangulation`: solve linear system from angle measurements; intersect bearing lines. |
| | Proximity | Grid-search cost minimization that penalizes mismatches between binary detections and candidate positions. |
| | Event-based (TOA) | Nonlinear least-squares fit of observed and predicted arrival times, assuming known wave speed. |
| Field variable prediction | Temporal imputation | Linear interpolation along the time axis. |
| | Forecasting | Simple extrapolation from most recent observations. |
| | Spatio-temporal imputation | Distance-weighted average of nearby spatial samples. |
| Tracking | Range-only | SCAAT-style EKF with custom Jacobians for range measurements. |
| | Bearing-only | EKF adapted for angular measurements. |
| | Range + bearing | Joint EKF fusing both modalities. |
| | Event-based (TOA) | Nonlinear least-squares estimation of position and event time from TOA data. |
| | Proximity | Geometric intersection of binary detection zones followed by convex-hull extraction. |
| Others | Tier 2 | No solution developed |
| | Tier 3 | No solution developed |

Table 7: Summary of first-principle baselines developed for each task family.

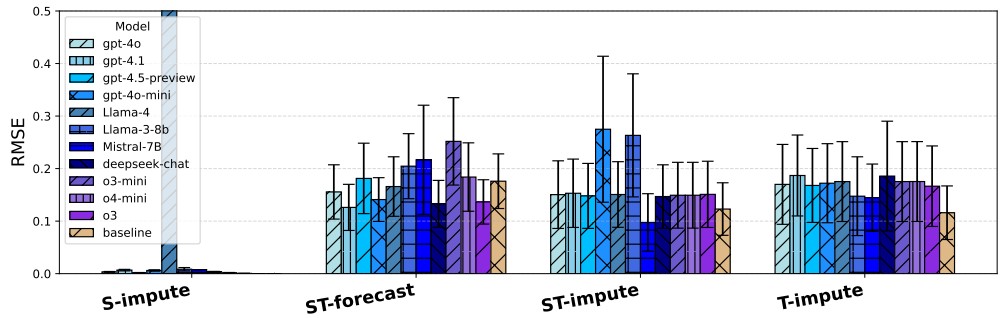

Figure 11: Field variable prediction compared with baselines (Llama-4 resorts to executing code, which is not allowed in the setting and results in high error).

ments. The EKF uses a non-linear measurement model with custom Jacobians to map predicted positions to sensor-measured distances. Similarly, for bearing-only tracking, we adapt the EKF framework to handle angular measurements. For range-bearing fusion, we extend the same EKF structure to jointly incorporate both modalities. The event-based baseline estimates both the spatial location and the event happening time using time-of-arrival (TOA) measurements from distributed sensors, formulating the problem as a nonlinear least-squares optimization. Lastly, for proximity-based tracking, we employ a geometric intersection method: binary detections (inside/outside a fixed radius) from multiple sensors are used to generate candidate regions, and a convex hull is applied to estimate the object's position from the intersecting detection zones.

**Others.** For POI identification and intent prediction based on user mobility trajectories, we design a finite state machine (FSM)-based baseline to provide a purely algorithmic, quantitative reference. This model captures sequential transition patterns from historical location data without relying on learning or world knowledge. For the remaining tasks, such as spatial relationships, direction, proximity, or temporal overlap, we do not construct first-principle baselines, as using spatial libraries (e.g., Shapely) and geospatial services (e.g., ArcGIS) yields perfect accuracy. Nonetheless, we would like to remind our readers that the random-guess baseline yields 50% accuracy due to a balanced distribution of positive and negative pairs. This design choice makes performance gains over the naive baseline a meaningful signal of reasoning ability.

## A.7 Field variable prediction compared with baselines (Figure 11)

## A.8 Results of solving problems by direct answering on STARK-S (Table 8)

Table 8 reports the exact benchmark results across a diverse set of spatiotemporal and navigation tasks, comparing general-purpose and open-source models. Overall, the strongest performance is concentrated among the LRMs (o3, o3-mini, o4-mini), which consistently achieve the lowest error rates across both field variable predictions, localization, tracking, and world-knowledge-aware reasoning. For example, o3 achieves top results in S-impute, T-loc-event, S-loc-event, and most tracking tasks, while o3-mini and o4-mini provide complementary strengths in ST-forecast and several localization subtasks. By contrast, general LLMs such as Llama-4, Llama-3-8b, and Mistral-7B underperform, particularly on tasks requiring fine-grained temporal or spatial reasoning

| model | gpt-4o | gpt-4.1 | gpt-4.5 | gpt-4o-mini | Llama-4 | deepseek | Llama-3-8b | Mistral-7B | o3-mini | o4-mini | o3 |
|---|---|---|---|---|---|---|---|---|---|---|---|
| S-impute | 0.004 | 0.005 | 0.001 | 0.005 | 0.986 | 0.002 | 0.005 | 0.820 | *0.001* | 0.000 | **0.000** |
| ST-forecast | 0.098 | 0.075 | 0.098 | 0.109 | 0.100 | *0.083* | 0.143 | **0.070** | 0.152 | 0.106 | 0.092 |
| ST-impute | 0.049 | 0.052 | *0.052* | 0.060 | 0.054 | 0.054 | 0.084 | **0.029** | 0.052 | 0.052 | 0.053 |
| T-impute | 0.054 | 0.080 | 0.063 | 0.063 | 0.063 | **0.044** | *0.054* | 0.129 | 0.063 | 0.063 | 0.049 |
| Loc-range | 1.792 | 1.172 | 1.260 | 3.042 | 0.563 | 1.809 | 3.403 | 2.737 | *0.042* | 0.039 | **0.036** |
| Loc-bearing | 2.993 | 2.514 | 0.610 | 3.082 | 1.124 | 2.299 | 3.206 | 2.828 | *0.133* | **0.117** | 0.126 |
| Loc-range-bearing | 1.716 | 1.695 | *0.056* | 2.117 | 0.059 | 0.056 | 2.982 | 2.518 | 0.057 | 0.058 | **0.043** |
| Loc-proximity | 1.140 | 0.980 | **0.915** | 1.197 | 1.264 | 1.217 | 1.872 | 2.385 | 1.090 | *1.006* | 1.283 |
| T-loc-event | 0.890 | 0.414 | 0.381 | 0.957 | 0.311 | 0.467 | 0.742 | 2.209 | 0.076 | *0.176* | **0.059** |
| S-loc-event | 2.336 | 2.398 | 1.902 | 2.426 | 2.708 | 2.989 | 2.987 | 2.547 | 0.198 | *0.258* | **0.177** |
| Track-range | 2.825 | 3.024 | 2.909 | 3.480 | 2.028 | 2.789 | 4.158 | 3.576 | *0.850* | 0.689 | **0.669** |
| Track-bearing | 2.991 | 3.268 | 1.793 | 3.064 | 1.959 | 1.524 | 4.014 | 3.664 | 0.638 | *0.764* | **0.580** |
| Track-range-bearing | 2.605 | 2.601 | 0.659 | 2.026 | 1.115 | *0.118* | 4.378 | 3.705 | **0.113** | 0.123 | 0.113 |
| Track-proximity | 1.984 | 2.112 | 2.095 | 2.195 | 2.216 | *1.857* | 2.884 | 2.804 | 2.161 | 1.853 | **1.798** |
| S-track-event | 3.917 | 3.652 | 4.204 | 3.775 | 3.313 | 3.742 | 4.217 | 3.523 | 1.542 | *1.830* | **1.052** |
| T-track-event | 0.414 | 0.184 | 0.142 | 0.184 | 1.066 | 0.306 | 0.882 | 0.841 | *0.094* | 0.068 | **0.059** |
| S relationship | 0.115 | 0.086 | 0.040 | 0.149 | *0.029* | 0.081 | 0.343 | 0.230 | **0.000** | 0.011 | **0.000** |
| T relationship | 0.062 | **0.000** | **0.000** | 0.077 | *0.046* | 0.031 | 0.523 | 0.231 | **0.000** | **0.000** | **0.000** |
| ST relationship | 0.119 | 0.087 | **0.008** | 0.183 | 0.198 | 0.143 | 0.508 | 0.460 | 0.024 | *0.032* | 0.024 |
| Landmark direction | *0.200* | *0.200* | *0.200* | 0.333 | **0.000** | 0.333 | 0.667 | 0.600 | 0.133 | **0.000** | **0.000** |
| Intent prediction | 0.200 | 0.143 | 0.200 | *0.200* | *0.133* | 0.200 | 0.600 | 0.600 | *0.133* | 0.067 | **0.000** |
| Landmark proximity | **0.000** | **0.000** | 0.067 | 0.600 | 0.533 | 0.067 | 0.467 | *0.200* | **0.000** | **0.000** | 0.067 |
| POI prediction | **0.000** | **0.000** | **0.000** | 0.600 | **0.000** | **0.000** | 0.333 | 0.600 | *0.200* | 0.067 | **0.000** |
| Route planning | **0.000** | **0.000** | **0.000** | **0.000** | **0.000** | **0.000** | **0.000** | 0.067 | **0.000** | **0.000** | **0.000** |
| ETA | **0.000** | **0.000** | **0.000** | **0.000** | **0.000** | **0.000** | **0.000** | 0.267 | **0.000** | **0.000** | **0.000** |
| Route segment | **0.000** | **0.000** | *0.067* | 0.033 | 0.133 | **0.000** | 0.133 | 0.250 | **0.000** | **0.000** | **0.000** |

Table 8: Benchmark results. Model performance measured by RMSPE (field variable prediction) and RMSE (others). Best results are **bold**, second best are underlined, and third best are *gray*.

## A.9 Results of solving problems by code interpreters on STARK-S (Table 9)

## A.10 Results of solving problems by direct answering on STARK-L (Table 10)

Table 10 reports the full-benchmark results after removing GPT-4.5, which was deprecated by OpenAI. The overall trend is consistent with the small-scale evaluation in Table 8: higher-capacity models such as GPT-4o, GPT-4.1, and o-series variants exhibit strong generalization across spatial, temporal, and spatiotemporal reasoning tasks, while smaller models (e.g., Llama-3/4-8B and Mistral-7B) struggle in localization and tracking subtasks. The relative ranking of models remains largely unchanged between STARK-S and STARK-L, confirming the stability of performance patterns across dataset scales.

## A.11 Task examples

### A.11.1 Tier 1

The following tables 18, 14, 15, 16, 17, as indicated by the title, show the task and sensor modality combinations.

### A.11.2 Tier 2

The following tables 19, 20, 21, as indicated by the title, show tier 2 examples..

## A.12 Tier 3

The following tables 22, 26, 23, 24, 28, 27, as indicated by the title, show tier 3 examples.

| model | gpt-4o | gpt-4.1 | gpt-4o-mini | deepseek-chat | o3-mini |
|---|---|---|---|---|---|
| S-impute | **0.000** | **0.000** | **0.000** | **0.000** | **0.000** |
| ST-forecast | 0.274 | 0.260 | 0.242 | *0.250* | **0.220** |
| ST-impute | 0.195 | 0.229 | 0.301 | *0.215* | **0.149** |
| T-impute | 0.175 | **0.137** | 0.232 | 0.187 | *0.175* |
| Loc-range | **0.124** | 0.217 | 0.750 | 0.750 | *0.548* |
| Loc-bearing | 116.756 | **0.124** | 2.087 | *0.137* | 0.133 |
| Loc-range-bearing | *0.132* | **0.059** | 0.163 | 0.131 | 0.217 |
| Loc-proximity | 1.218 | **1.019** | 1.133 | 1.061 | *1.085* |
| T-loc-event | 16.712 | **0.021** | *0.624* | 73.490 | 0.032 |
| S-loc-event | 2.000 | 0.083 | 4.345 | *0.144* | **0.083** |
| Track-range | 3.249 | *2.994* | 15.854 | 2.841 | **1.945** |
| Track-bearing | 66.386 | *1.867* | 3.737 | 1.434 | **1.128** |
| Track-range-bearing | 2.350 | *2.854* | 3.901 | 3.521 | **0.220** |
| Track-proximity | 2.302 | 2.004 | *2.161* | 2.376 | **1.953** |
| S-track-event | 24.094 | 1.839 | 64.045 | *6.431* | **1.823** |
| T-track-event | 0.493 | 0.743 | *0.728* | **0.154** | 10.489 |
| S relationship | **0.000** | 0.034 | **0.000** | **0.000** | **0.000** |
| T relationship | 0.043 | *0.051* | 0.094 | 0.043 | **0.000** |
| ST relationship | 0.206 | *0.202* | 0.242 | 0.177 | **0.127** |
| Landmark direction | 0.360 | 0.400 | 0.208 | *0.280* | **0.200** |
| Intent pred. | *0.440* | *0.440* | 0.520 | 0.280 | **0.240** |
| Landmark proximity | *0.520* | 0.440 | 0.760 | 0.720 | **0.240** |
| POI pred. | 0.160 | *0.280* | 0.320 | **0.120** | 0.160 |
| Route planning | **0.000** | **0.000** | 0.167 | **0.000** | *0.200* |
| ETA calc. | 0.040 | *0.080* | **0.000** | *0.080* | *0.080* |
| Route segment dur. | 0.560 | *0.280* | 0.240 | **0.200** | **0.200** |

Table 9: Benchmark results (coding). The table shows the absolute performance of solving problems with Python coding.

| model | gpt-4o | gpt-4.1 | gpt-4o-mini | Llama-4 | deepseek-chat | Llama-3-8b | Mistral-7B | o3-mini | o4-mini | o3 |
|---|---|---|---|---|---|---|---|---|---|---|
| S-impute | 0.117 | 0.008 | 0.008 | 0.893 | 0.003 | 0.013 | 0.842 | 0.001 | *0.001* | **0.000** |
| ST-forecast | *0.147* | 0.147 | 0.164 | 0.166 | **0.141** | 0.176 | 0.199 | 0.158 | 0.161 | 0.173 |
| ST-impute | 0.101 | 0.104 | 0.127 | 0.108 | 0.105 | 0.143 | 0.173 | *0.102* | **0.101** | 0.116 |
| T-impute | 0.073 | 0.069 | 0.079 | **0.067** | 0.076 | 0.082 | 0.132 | 0.070 | *0.070* | 0.087 |
| Loc-range | 2.177 | 1.562 | 2.812 | 0.493 | 2.080 | 2.962 | 2.743 | *0.056* | 0.044 | **0.043** |
| Loc-bearing | 2.895 | 2.620 | 2.926 | 1.688 | 2.223 | 3.032 | 2.813 | *0.166* | **0.151** | 0.153 |
| Loc-range-bearing | 1.855 | 1.645 | 1.932 | 0.116 | 0.119 | 3.031 | 2.851 | 0.062 | *0.063* | **0.053** |
| Loc-proximity | 1.065 | 1.034 | 1.054 | 1.034 | 0.958 | 1.413 | 1.801 | *0.956* | 0.878 | **0.858** |
| T-loc-event | 1.334 | 0.638 | 0.833 | 0.398 | 0.389 | 1.171 | 2.229 | 0.153 | *0.191* | **0.064** |
| S-loc-event | 2.790 | 2.584 | 2.890 | 2.746 | 2.772 | 2.924 | 2.765 | 0.292 | *0.593* | **0.184** |
| Track-range | 2.891 | 2.934 | 3.610 | 2.141 | 2.642 | 4.099 | 3.789 | 0.656 | *0.852* | **0.556** |
| Track-bearing | 3.644 | 3.410 | 3.628 | 2.687 | 2.090 | 4.272 | 3.611 | 0.822 | *0.866* | **0.791** |
| Track-range-bearing | 1.179 | 2.048 | 1.357 | 0.836 | 0.138 | 4.151 | 3.820 | *0.128* | 0.124 | **0.119** |
| Track-proximity | **2.020** | 2.024 | 2.097 | 2.209 | 2.156 | 2.657 | 2.763 | 2.276 | 2.205 | *2.083* |
| S-track-event | 3.340 | 3.750 | 3.619 | 3.493 | 3.647 | 4.040 | 3.488 | 1.265 | *1.815* | **0.931** |
| T-track-event | 0.329 | *0.136* | 0.255 | 0.447 | 0.141 | 0.383 | 0.428 | 0.114 | 0.138 | **0.080** |
| S relationship | 0.074 | *0.053* | 0.169 | 0.064 | 0.069 | 0.334 | 0.311 | **0.000** | 0.009 | **0.000** |
| T relationship | **0.000** | **0.000** | 0.021 | **0.000** | **0.000** | 0.467 | *0.363* | **0.000** | **0.000** | **0.000** |
| ST relationship | 0.133 | *0.042* | 0.187 | 0.214 | 0.130 | 0.434 | 0.507 | **0.026** | 0.029 | 0.057 |
| Landmark direction | 0.259 | 0.383 | 0.296 | 0.400 | 0.321 | 0.481 | 0.617 | *0.150* | 0.074 | **0.037** |
| Intent prediction | 0.207 | 0.332 | 0.398 | 0.253 | 0.237 | 0.506 | 0.361 | 0.282 | *0.224* | **0.195** |
| Landmark proximity | *0.282* | 0.266 | 0.577 | 0.552 | 0.394 | 0.419 | 0.481 | 0.307 | **0.216** | 0.303 |
| POI prediction | 0.162 | *0.158* | 0.465 | 0.154 | 0.249 | 0.423 | 0.527 | 0.212 | 0.199 | **0.124** |
| Route planning | **0.000** | **0.000** | 0.136 | **0.000** | **0.000** | **0.000** | 0.333 | *0.148* | **0.000** | **0.000** |
| ETA | 0.020 | *0.030* | 0.139 | 0.099 | 0.020 | 0.238 | 0.475 | *0.030* | **0.000** | **0.000** |
| Route segment dur. | 0.136 | *0.099* | 0.148 | 0.148 | 0.086 | 0.531 | 0.593 | 0.173 | 0.185 | **0.074** |

Table 10: Benchmark results. Model performance measured by RMSPE (field variable prediction) and RMSE (others). Best results are **bold**, second best are underlined, and third best are *gray*.

| Examples | Help me determine the location of an object in a 2D plane based on sensor measurements. |
|---|---|
| | Return the location of the object in the following format: |
| | [RESULTS_START] [x, y] [RESULTS_END] |
| | Given Information: There are four sensors, each providing a range-only (distance) measurement to the object. These measurements indicate the distance from the object to the sensors. The sensor locations in Cartesian coordinates ([x, y]) are: |
| | - Sensor A: [9.18, 4.97] - Sensor B: [0.11, 6.92] - Sensor C: [8.64, 1.80] - Sensor D: [3.55, 5.54] |
| | The sensor readings are: 8.4643, 1.0005, 9.2132, 2.9811. - The reading represents the distance from the object to each sensor. - The measurements may contain noise. |
| | Question: |
| | Based on the given data, where is the object most likely located? Provide the estimated [x, y] coordinates. |
| | Do not put text between the keywords [RESULTS_START] and [RESULTS_END]. |
| Modality | Range |
| Answer | [1.0750516142060274, 7.2038399470332815] |

Table 11: Spatial localization-Range

| Examples | Help me determine the location of an object in a 2D plane based on sensor measurements. |
|---|---|
| | Return the location of the object in the following format: |
| | [RESULTS_START] [x, y] [RESULTS_END] |
| | Do not include any additional text between [RESULTS_START] and [RESULTS_END]. |
| | Given Information: There are four sensors, each providing: 1. A range measurement (distance) from the sensor to the object. 2. A bearing measurement (angle) indicating the direction of the object relative to the sensor. The sensor locations in Cartesian coordinates ([x, y]) are: |
| | - Sensor A: [1.09, 2.86] - Sensor B: [7.03, 1.40] - Sensor C: [9.34, 0.91] - Sensor D: [4.62, 3.05] |
| | The sensor readings are: [8.4303, 347.2164], [2.2862, 352.0771], [0.0185, 131.7104], [5.0572, 336.6944]. - Each reading is in the form [A, B], where: - A represents the distance from the object to the sensor. - B represents the bearing angle (in degrees), measured within the range [0, 360). i) Angle unit: degree, within the range [0, 360). ii) The bearing angle for each sensor is measured from the sensor's own position (treated as the origin for its measurement). iii) The reference direction is the positive x-axis (a horizontal ray extending to the right from the sensor's location). iv) Angles increase counterclockwise (CCW) from this reference direction. - The measurements may contain noise. |
| | Question: |
| | Using the given range and bearing measurements, determine the most likely location of the object in Cartesian coordinates [x, y]. |
| Modality | Bearing |
| Answer | [1.263400029932652, 0.6843345416596847] |

Table 12: Spatial localization-bearing

| Examples | Help me determine the location of an object in a 2D plane based on sensor measurements.
Return the location of the object in the following format:
[RESULTS_START] [x, y] [RESULTS_END]
Do not include any additional text between [RESULTS_START] and [RESULTS_END].
Given Information:
There are four sensors, each providing a region-based measurement indicating whether the object is within a defined detection region. Each sensor provides a binary reading: - 1: The object is inside the sensor detection region. - 0: The object is outside the sensor detection region. The sensor locations in Cartesian coordinates ([x, y]) are:
- Sensor A: [4.21, 8.68] - Sensor B: [8.50, 3.99] - Sensor C: [4.47, 9.16] - Sensor D: [0.15, 4.68]
Each sensor detects objects within a disk of radius 4.0 around its location
The sensor readings are: 1.0000, 1.0000, 1.0000, 0.0000 - Each reading is in the form [A], where: - A is either 1 or 0, indicating whether the object is within the sensor detection region. - The measurements may contain noise.
Question:
Using the given region-based sensor measurements, determine the most likely location of the object in Cartesian coordinates [x, y]. |
|---|---|
| Modality | Proximity |
| Answer | [6.93892999169199, 6.201391778478937] |

Table 13: Spatial localization-Proximity

| Examples | You have a 2D region (for simplicity, a 10 km by 10 km square) in which a seismic event (like a small earthquake or underground explosion) occurs at an unknown location. A set of seismic sensors (geophones) is placed in this region at known fixed positions. Each sensor is event-based: it records the time when it detects the seismic wave.
Detection-Time Model: Suppose each sensor's detection time T depends linearly on the distance d from the event, using the seismic wave speed (5km/s):
T = d/5 (second),
meaning that if the sensor is 10 kilometers away from the event, it will detect it at 2 seconds after ignition, if it is 5 kilometers away, it detects at 1 second, etc.
Given Information:
There are four sensors, each providing the time that an event is detected. Each sensor provides the detection of the event in second: The sensor locations in Cartesian coordinates ([x_i, y_i], in km) are:
- Sensor A: [2.01, 2.80] - Sensor B: [3.21, 9.68] - Sensor C: [8.06, 4.07] - Sensor D: [2.72, 0.73]
The sensor readings are: 4.1103, 5.3980, 4.7342, 3.7564 - Each reading is in the form [A], where: - A a positive number, indicating the second the event is detected. - The measurements may contain noise. - Suppose the event happened at time t. The event's occurrence time is related to sensor readings by: A = t + T.
Localization Goal: By collecting these detection times from multiple sensors, the objective is to estimate where the seismic event occurred (in km). Determine the location in the required format [RESULTS_START] [x, y] [RESULTS_END]. |
|---|---|
| Modality | Event |
| Answer | [3.4805565559661114, 0.7611464626957665] |

Table 14: Spatial localization-Event

| | |
|---|---|
| Examples | Help me determine the time in seconds when a seismic event occurred in a 2D area based on sensor measurements.
Format Requirement:
Return the estimated time only in the following format, replacing t with the computed value:
[RESULTS_START] [t] [RESULTS_END]
Do not include any additional text between [RESULTS_START] and [RESULTS_END].
Scenario: You have a 2D region (for simplicity, a 10 km by 10 km square) in which a seismic event (like a small earthquake or underground explosion) occurs at an unknown location. A set of seismic sensors (geophones) is placed in this region at known fixed positions. Each sensor is event-based: it records the time when it detects the seismic wave.
Detection-Time Model: Suppose each sensor's detection time T depends linearly on the distance d from the event, using the seismic wave speed (5km/s):
$T = d/5$ (second),
meaning that if the sensor is 10 kilometers away from the event, it will detect it at 2 seconds after ignition, if it is 5 kilometers away, it detects at 1 second, etc.
Given Information:
There are four sensors, each providing the time that an event is detected. Each sensor provides the detection of the event in second: The sensor locations in Cartesian coordinates ([x, y], in km) are:
- Sensor A: [4.11, 3.28] - Sensor B: [4.06, 7.63] - Sensor C: [4.80, 5.48] - Sensor D: [2.51, 1.51]
The sensor readings are: 5.0332, 4.1634, 4.6190, 5.4055 - Each reading is in the form [A], where: - A a positive number, indicating the **second** the event is detected. - The measurements may contain noise. - The event's occurrence time t is related to sensor readings by: $A = t + T$.
Localization Goal: By collecting these detection times in **second** from multiple sensors, the objective is to estimate when the seismic event occurred in **second**. Determine the time in the required format [RESULTS_START] [t] [RESULTS_END]. |
| Modality | Event |
| Answer | [3.961012243187861, 3.961012243187861] |

Table 15: Temporal localization-Event

## A.13 Reasoning over spatiotemporal relationship

Table 29 highlights the class-wise performance difference between LLMs and LRMs in spatiotemporal relationship reasoning tasks. LLMs such as GPT-4o and GPT-4.1 struggle more in line-line intersection (L-L-intersects) and line-polygon intersection (L-Pg-intersects), where their performance decreases compared to other tasks. Similarly, for temporal relationship reasoning, the LLMs find it more challenging to handle the overlaps-with task. In contrast, LRMs such as o3-mini and o3 exhibit robust performance across all evaluated categories, consistently achieving superior accuracy or near-perfect results. This demonstrates the effectiveness and specialized capability of LRMs in compound spatiotemporal reasoning scenarios. See Appendix A.14 and A.15 for spatial relationship and temporal relationship results.

## A.14 Spatial relationship (Table 30)

## A.15 Temporal relationship (Table 31)

| Examples | Help me track the location of an object in a 2D plane at each step based on temporal sensor measurements. |
|---|---|
| | Given Information: You will receive a set of range-only measurements from up to four sensors at each time stamp. These measurements indicate the distance from the object to the sensors. The sensor locations in Cartesian coordinates ([x, y]) are: |
| | - Sensor A: [3.37, 4.61] - Sensor B: [7.43, 4.62] - Sensor C: [4.89, 5.52] - Sensor D: [0.59, 6.46] |
| | The sensor readings are an array. - The reading represents the distance from the object to each sensor. - The measurements may contain noise. - Some sensors may not report a distance (i.e., a missing reading) at certain time stamps. |
| | Objective: Based on the available distance measurements at each time stamp (including the historical data), estimate the most likely [x, y] coordinates of the object in the 2D plane. |
| | Output Format: Your response **must** include the object's estimated location in the following format: |
| | [RESULTS_START] [x, y] [RESULTS_END] |
| | Do not include any text or additional formatting between the [RESULTS_START] and [RESULTS_END]. |
| | Example: Suppose at a certain time stamp, sensor 1 and sensor 3 provide distance measurements, while sensors 2 and 4 do not. Use only the available readings and historical data to compute the object's best-guess location. |
| | Return the results in the exact format: |
| | [RESULTS_START] [1.23, 4.56] [RESULTS_END] |
| | Question: |
| | Based on the given data, where is the object most likely located at each step? Provide the estimated [x, y] coordinates. You **must** give an estimation. |
| Modality | Range |
| Question #1 | Step 0 | At time 0.0000, the sensor readings are 6.3630, 9.1967, nan, nan |
| Answer #1 | [0.0, 10.0] |
| Question #2 | Step 0 | At time 0.0000, the sensor readings are 6.3630, 9.1967, nan, nan |
| Answer #2 | [0.2394074101698792, 9.94759324595609] |
| Question #3 | Step 1 | At time 0.2041, the sensor readings are nan, 8.9666, nan, nan |
| Answer #3 | [1.263400029932652, 0.6843345416596847] |
| ... | ... |
| Question #9 | Step 8 | At time 1.6327, the sensor readings are nan, 7.1842, nan, nan |
| Answer #9 | [1.4291178779723746, 9.046650349589203] |
| Question #10 | Step 9 | At time 1.8367, the sensor readings are nan, 6.7058, 4.1855, nan |
| Answer #10 | [1.6194253467978819, 8.752139659211627] |

Table 16: Spatial tracking-Range. The model is required to estimate the new object location when new sensor readings come.

| Examples | There is a shooter moving in a 2D plane (10 x 10 km) monitored by sensors. Help me track the time a gunshot occurred in a 2D plane (10 x 10 km) based on time-of-arrival (TOA) measurements from up to four microphones. |
|---|---|
| | Given Information: At each step, you will receive a set of time-of-arrival measurements from up to four sensors at each time stamp (in minute). These measurements indicate how long after the shot was fired that each sensor detected the sound. The sensor locations in Cartesian coordinates ([x, y], in km) are: |
| | - Sensor A: [1.54, 8.20] - Sensor B: [5.51, 6.43] - Sensor C: [4.37, 9.95] - Sensor D: [5.04, 0.39] |
| | But for the purpose of this problem, we are focusing on estimating the shot's initial firing time (T_0). The sensor readings come in an array: - The reading represents the TOA at the corresponding sensor location. - Some sensors may not report a reading (i.e., missing data) at certain time stamps. - The measurements may contain noise. |
| | Objective: Based on the available time-of-arrival measurements at each time stamp (including historical data), estimate the most likely time $T_0$ (in min) at which the shot was fired. |
| | Output Format: Your response **must** include the shooter's estimated location in the following format: |
| | [RESULTS_START] [T_0] [RESULTS_END] |
| | Do not include any text or additional formatting between the [RESULTS_START] and [RESULTS_END]. |
| | Acoustic-Time Model: Assume each microphone receives the shot at time T given by: $T = d / v + T_0$, |
| | where d = distance from the shooter to the microphone (need to be estimated), v = speed of sound (approximately 20 km/min), $T_0$ = the time offset when the shot was actually fired (which you may approximate or fit from the data). |
| | Example: Suppose at a certain time stamp, sensor 1 and sensor 3 provide TOA measurements, while sensors 2 and 4 do not. Use only the available readings and historical data to compute the best-guess firing time. |
| | Return the result in the exact format (in min) at each step: |
| | [RESULTS_START] [0.1234] [RESULTS_END] |
| | Question: |
| | Based on the given TOA data, estimate the $T_0$ (in min) when the shot was fired at each step. Provide a numeric answer and you **must** give an estimation. |

| Modality | Range |
|---|---|
| Question #1 | Step 0 | The corresponding sensor readings are 9.5777, 9.4917, 9.6516, 9.2405 |
| Answer #1 | [9.160943872637121] |
| Question #2 | Step 1 | The corresponding sensor readings are 9.8002, 9.6949, 9.8543, 9.4018 |
| Answer #2 | [9.365025505290182] |
| Question #3 | Step 2 | The corresponding sensor readings are 10.0176, 9.8928, 10.0620, 9.5798 |
| Answer #3 | [9.569107137943243] |
| ... | ... |
| Question #9 | Step 8 | The corresponding sensor readings are 11.2803, 11.0751, nan, 10.9987 |
| Answer #9 | [10.793596933861611] |
| Question #10 | Step 9 | The corresponding sensor readings are 11.4759, 11.2622, 11.4417, 11.2287 |
| Answer #10 | [10.997678566514672] |

Table 17: Temporal tracking-Range. The model is required to estimate the event time when new sensor readings come.

| Examples | Help me answer a question involving forecasting of sensory data from a sensor network. You will be given a sequence of sensor readings from a group of sensor devices which are located in proximity to each other, and where each reading is taken at consecutive time steps. Your objective is to predict the next value for all sensors at the next time step. Some more context: this sensory network is deployed at various locations in a city to measure PM2.5 air quality readings. These readings have been sampled from January 7th 2025 to January 30th from various locations around Los Angeles. During this time, there were a few significant wildfires occuring. The readings are taken at a frequency of 1/hour. Sensor values from previous time steps: 
 What is the value of the group's sensor readings at time t? Note that you will need to provide several values (for each sensor in the group) 
 Sensor values from previous time steps : 
 Sensor readings taken from time t-7 (real world time is January 11, 2025 at 10:00:02 AM): Sensor located at Lat: 33.83253, Long: -118.18794 has value: 50.9 Sensor located at Lat: 33.855965, Long: -118.28898 has value: 0.0 Sensor located at Lat: 33.880737, Long: -118.38882 has value: 26.0 
 Sensor readings taken from time t-6 (real world time is January 11, 2025 at 11:00:02 AM): Sensor located at Lat: 33.83253, Long: -118.18794 has value: 39.2 Sensor located at Lat: 33.855965, Long: -118.28898 has value: 1.0 Sensor located at Lat: 33.880737, Long: -118.38882 has value: 12.2 
 Sensor readings taken from time t-5 (real world time is January 11, 2025 at 12:00:02 PM): Sensor located at Lat: 33.83253, Long: -118.18794 has value: 43.6 Sensor located at Lat: 33.855965, Long: -118.28898 has value: 1.1 Sensor located at Lat: 33.880737, Long: -118.38882 has value: 10.7 
 Sensor readings taken from time t-4 (real world time is January 11, 2025 at 01:00:02 PM): Sensor located at Lat: 33.83253, Long: -118.18794 has value: 29.6 Sensor located at Lat: 33.855965, Long: -118.28898 has value: 0.0 Sensor located at Lat: 33.880737, Long: -118.38882 has value: 13.3 
 Sensor readings taken from time t-3 (real world time is January 11, 2025 at 02:00:02 PM): Sensor located at Lat: 33.83253, Long: -118.18794 has value: 12.6 Sensor located at Lat: 33.855965, Long: -118.28898 has value: 0.5 Sensor located at Lat: 33.880737, Long: -118.38882 has value: 14.8 
 Sensor readings taken from time t-2 (real world time is January 11, 2025 at 03:00:02 PM): Sensor located at Lat: 33.83253, Long: -118.18794 has value: 13.8 Sensor located at Lat: 33.855965, Long: -118.28898 has value: 0.0 Sensor located at Lat: 33.880737, Long: -118.38882 has value: 9.3 
 Sensor readings taken from time t-1 (real world time is January 11, 2025 at 04:00:02 PM): Sensor located at Lat: 33.83253, Long: -118.18794 has value: 17.1 Sensor located at Lat: 33.855965, Long: -118.28898 has value: 1.2 Sensor located at Lat: 33.880737, Long: -118.38882 has value: 9.3 What is the value of the group's sensor reading at time t (real world time is January 11, 2025 at 05:00:02 PM)? Note that you will need to provide several values (for each sensor in the group) 
 Please respond with your output as a list of predicted values in the following format: [RESULT_START] [predictions] [RESULT_END] Do not include any text or additional formatting between the [RESULTS_START] and [RESULTS_END]. Example: [RESULTS_START] [23.5, 35.7, 45.3, 75.2, 24.2] [RESULTS_END] |
|---|---|
| Answer | [32.8, 0.2, 21.4] |

Table 18: Spatiotemporal Forecast

| | |
|---|---|
| Examples | Help me answer question regarding spatial relationship in a 2D plane: |
| | Given Information: You will be provided with geometric information involving three types of 2D geometries—Point, LineString, and Polygon—all defined using the ESRI (Environmental Systems Research Institute) geometric format. These geometries are expressed as lists of coordinates in a Cartesian plane. |
| | Point: A single coordinate location in space, defined as a tuple: |
| | $[(x, y)]$. |
| | LineString: A sequence of points that forms a continuous line. It is represented as an ordered list of coordinate pairs: |
| | $[(x_1, y_1), (x_2, y_2), ...(x_n, y_n)]$. |
| | Polygon: A closed shape formed by a sequence of coordinate pairs where the first and last points are the same to close the loop: |
| | $[(x_1, y_1), (x_2, y_2), ...(x_n, y_n), (x_1, y_1)]$. |
| | Spatial Relationships (based on ArcGIS model) is as follows. During computation, you should account for floating-point precision with a numerical tolerance. For instance, 'Equals' should return True not just for exact mathematical identity, but if two geometries are identical within this tolerance. |
| | ArcGIS defines spatial relationships as logical conditions between geometric objects: 1. Equals: Returns True if two geometries represent the same shape and location. 2. Intersects: Returns True if the geometries share any portion of space, including edges or points. 3. Contains: Returns True if one geometry completely encloses another. 4. Within: The reverse of Contains. Returns True if the first geometry lies completely inside the second. 5. Crosses: Returns True if the geometries intersect in a way where they share some interior points but are of different dimensions (e.g., a line crossing another line or a line crossing a polygon). 6. Touches: Returns True if the geometries share only a boundary or a point but no interior space. 7. Overlaps: Returns True if the geometries share some, but not all, interior points, and are of the same dimension. |
| | Objective: |
| | Determine whether the Linestring [(-0.6139, 9.0467), (3.4341, 9.0467)] has the spatial relationship **intersects** with the Polygon [(0.3861, 8.8963), (0.8963, 8.1375), (1.8078, 8.0632), (2.4341, 8.7295), (2.3036, 9.6346), (1.5147, 10.0970), (0.6613, 9.7684), (0.3861, 8.8963)]? |
| | Answer 1 if answer is Yes. Otherwise, answer 0. |
| | Output Format: Your response **must** include the answer (0 or 1) in the following format: |
| | [RESULTS_START] [p] [RESULT_END] |
| | You may include explanatory text elsewhere in your response. However, do not include any text or additional formatting between [RESULTS_START] and [RESULTS_END]. |
| | Example: |
| | [RESULTS_START] [1] [RESULT_END] |
| Modality | Range |
| Answer | [1] |

Table 19: Spatial relationship

| Examples | Help me answer question regarding temporal relationship: |
|---|---|
| | Given Information: You will be provided with intervals defined by $(x_i, y_i)$. |
| | $x_i$ and $x_i$ are non negative numbers. |
| | Temporal Relationships (based on Allen's interval algebra): |
| | 1. Before (A before B): A ends before B starts. 2. After (A after B): A starts after B ends. 3. Meets (A meets B): A ends exactly when B starts. 4. Met-By (A met-by B): A starts exactly when B ends. 5. Overlaps (A overlaps B): A starts before B starts, and ends after B starts but before B ends. 6. Overlapped-By (A overlapped-by B): A starts after B starts, and ends after B ends but before A ends. 7. Starts (A starts B): A and B start at the same time, but A ends before B ends. 8. Started-By (A started-by B): A and B start at the same time, but A ends after B ends. 9. During (A during B): A starts after B starts and ends before B ends. 10. Contains (A contains B): A starts before B starts and ends after B ends. 11. Finishes (A finishes B): A and B end at the same time, but A starts after B starts. 12. Finished-By (A finished-by B): A and B end at the same time, but A starts before B starts. 13. Equals (A equals B): A and B start and end at the same time. |
| | Objective: |
| | Determine whether the time interval (3.7558, 6.4075) has the temporal relationship **overlaps with** with the time interval (39.8870, 44.8131)? |
| | Answer 1 if answer is Yes. Otherwise, answer 0. |
| | Output Format: Your response **must** include the answer (0 or 1) in the following format: |
| | [RESULTS_START] [p] [RESULTS_END] |
| | You may include explanatory text elsewhere in your response. However, do not include any text or additional formatting between [RESULTS_START] and [RESULTS_END]. |
| | Example: |
| | [RESULTS_START] [1] [RESULTS_END] |
| Modality | Range |
| Answer | [0] |

Table 20: Temporal relationship

| | |
|---|---|
| Examples | Help me answer question regarding spatial relationship in a 2D plane:
Given Information:
You will receive a series of object trajectory and the corresponding timestamps of the coordinates in the trajectory. You can treat the trajectory as linestring.
Sensor A: $[(x_1, y_1), (x_2, y_2), ..., (x_n, y_n)]$
Timestamp: $[t1, t2, ..., tn]$
You will be provided with geometric information involving three types of 2D geometries—Point, LineString, and Polygon—all defined using the ESRI (Environmental Systems Research Institute) geometric format. These geometries are expressed as lists of coordinates in a Cartesian plane.
Point: A single coordinate location in space, defined as a tuple:
$[(x, y)]$.
LineString: A sequence of points that forms a continuous line. It is represented as an ordered list of coordinate pairs:
$[(x_1, y_1), (x_2, y2), ...(x_n, y_n)]$.
Polygon: A closed shape formed by a sequence of coordinate pairs where the first and last points are the same to close the loop:
$[(x_1, y_1), (x_2, y2), ...(x_n, y_n), (x_1, y_1)]$.
Spatial Relationships (based on ArcGIS model) are as follows. During computation, you should account for floating-point precision with a numerical tolerance. For instance, 'Equals' should return True not just for exact mathematical identity, but if two geometries are identical within this tolerance.
ArcGIS defines spatial relationships as logical conditions between geometric objects:
...
Temporal Relationships (based on Allen's interval algebra):
...
Objective:
Determine whether the time interval during which the EVENT holds has the temporal relationship **is_equal_to** with the reference interval (6.9569, 9.5423)? EVENT: the following object trajectory has the spatial relationship **intersects** with Polygon [(8.9995, 9.6963), (9.4793, 9.4669), (9.9179, 9.7677), (9.8767, 10.2979), (9.3969, 10.5273), (8.9583, 10.2265), (8.9995, 9.6963)]
For any interaction between a trajectory (you can view it as a LineString) and a fixed geometry—whether that geometry is another LineString, a Point, or a Polygon—define the "event interval" as follows:
1. Pick a trajectory segment and a predicate. 2. Project the segment endpoints back to their timestamps. For each contiguous satisfying segment, let $t_1$ be the time at the segment's first vertex and $t_2$ be the time at its last vertex. 3. Define the event interval as the union of all **$[t_1, t_2]$** intervals in which the predicate is true. 4. Do not include portions of the trajectory before the relationship begins or after it ends. Do not interpolate.
Answer 1 if answer is Yes. Otherwise, answer 0.
Object trajectory: [(10.0000, 9.6138), (9.8171, 9.8075), (9.6491, 9.9374), (9.4950, 10.0000), (9.4381, 9.9971), (9.2219, 9.9145), (9.0990, 9.7653), (8.9822, 9.5463), (8.8692, 9.2601), (8.7575, 8.9104)] Timestamp: [1.9871, 3.0075, 4.0279, 5.0483, 6.0687, 7.0891, 8.1095, 9.1299, 10.1503, 11.1707]
Output Format: Your response **must** include the answer (0 or 1) in the following format:
[RESULTS_START] [p] [RESULTS_END]
You may include explanatory text elsewhere in your response. However, do not include any text or additional formatting between [RESULTS_START] and [RESULTS_END].
Example:
[RESULTS_START] [1] [RESULTS_END] |
| Answer | [0] |

Table 21: Spatioemporal relationship

| Example | Determine the most accurate spatial relationship between Statue of Liberty, New York, NY and Empire State Building, New York, NY is north-west of, selecting from the options: ['north of', 'north-east of', 'east of', 'south-east of', 'south of', 'south-west of', 'west of', 'north-west of']
Answer 1 if answer is Yes. Otherwise, answer 0. |
|---|---|
| Answer | [0] |

Table 22: Landmark direction

| Example | Is there a hospital within 200 metres of Santa Monica Pier, Santa Monica, CA?
Answer 1 if answer is Yes. Otherwise, answer 0. |
|---|---|
| Answer | [0] |

Table 23: Landmark proximity

| Example | Help me to answer the following question regarding spatio-temporal reasoning:
Given information:
1. Start at Lombard Street, San Francisco, CA 2. Go northeast on Lombard St toward Leavenworth St 3. At the stop sign, turn left on Leavenworth St 4. At the stop sign, turn left on Chestnut St 5. At the stop sign, turn left on Larkin St 6. At the stop sign, turn right on Lombard St 7. Take exit 442 on the right toward Alexander Avenue 8. Make a sharp left on Sausalito Lateral toward US-101 S / San Francisco 9. Merge onto Golden Gate Brg S (US-101 S) 10. Finish at Golden Gate Bridge, San Francisco, CA, on the right
Objective:
Does the route from Lombard Street, San Francisco, CA to Golden Gate Bridge, San Francisco, CA pass by San Francisco International Airport?
Answer 1 if answer is Yes. Otherwise, answer 0. |
|---|---|
| Answer | [0] |

Table 24: Route planning

| Example | If I leave Los Angeles International Airport by car at 9:12 AM, can I arrive at Santa Monica Pier, Santa Monica, CA by 9:26 AM? You can assume an average travel speed 47.8259 km/h. Answer directly based on the reasoning of spatial and/or temporal information.
Answer 1 if answer is Yes. Otherwise, answer 0. |
|---|---|
| Answer | [0] |

Table 25: ETA calculation

| Example | I drove from Fisherman's Wharf, San Francisco, CA to Golden Gate Bridge, San Francisco, CA starting at 10:00 AM, and the trip took about 19 minutes. Was I likely to witness the accident that occurred along the route from (Bay St, San Francisco, CA, 94109, USA) to (Mile 9.2 Us Hwy 101 N, San Francisco, CA, 94129, USA) between 9:42 AM and 9:57 AM? Answer directly based on the reasoning of spatial and/or temporal information.
Answer 1 if answer is Yes. Otherwise, answer 0. |
|---|---|
| Answer | [0] |

Table 26: Route segment duration

| | |
|---|---|
| Example | Given Information:
The coordinates of locations are as follows:
Location 0 coordiante: (34.051645489083, -118.243997724533).
Location 1 coordiante: (34.018319580967, -118.492826418854).
Location 2 coordinate: (34.050016, -118.261176).
Location 3 coordinate: (34.033664, -118.229166).
Location 4 coordinate: (34.072917, -118.35729).
Location 5 coordinate: (34.10156947286, -118.340960963788).
Location 6 coordinate: (33.99535, -118.475103).
Location 7 coordinate: (34.132390527116, -118.280364162419).
Location 8 coordinate: (34.043871172843, -118.266447633989).
The user's trajectory and information of each day are as follows:
The date is 2023-11-06. Monday. Today's weather is sunny, and air quality is moderate. From 07:00 to 09:00 traffic is busy, from 16:00 to 18:00 traffic is busy. From 13:00 to 14:00 the user feel fatigued, at 16:00 the user feel fatigued, from 19:00 to 21:00 the user feel fatigued. Today there is a popular Laker game. Starting from 16:00 on date 2023-11-06, the user's hourly trajectory is: 2, 0, 8, 2, 2, 2, 2, 0.
The date is 2023-11-07. Tuesday. Today's weather is sunny, and air quality is unhealthy. From 07:00 to 09:00 traffic is busy, from 16:00 to 18:00 traffic is busy. From 19:00 to 21:00 the user feel fatigued. A popular movie premieres today. Starting from 00:00 on date 2023-11-07, the user's hourly trajectory is: 0, 0, 0, 0, 0, 0, 7, 5, 5, 5, 5, 5, 5, 3, 3, 2, 2, 2, 2, 3, 5, 5, 5, 0.
The date is 2023-11-08. Wednesday. Today's weather is sunny, and air quality is good. From 07:00 to 09:00 traffic is busy, from 16:00 to 18:00 traffic is busy. At 14:00 the user feel fatigued, from 19:00 to 21:00 the user feel fatigued. The user has planned a date night. Starting from 00:00 on date 2023-11-08, the user's hourly trajectory is: 0, 0, 0, 0, 0, 0, 2, 2, 2, 2, 6, 6, 3, 3, 5, 7, 7, 5, 5, 5, 5, 0, 0, 0.
... ...
The date is 2023-11-12. Sunday. Today's weather is cloudy, and air quality is good. From 12:00 to 18:00 traffic is busy. From 19:00 to 21:00 the user feel fatigued. Today there is a big sales event, like Black Friday. Starting from 00:00 on date 2023-11-12, the user's hourly trajectory is: 0, 0, 0, 0, 0, 0, 6, 6, 0, 4, 4, 4, 4, 3, 4, 4, 4, 4, 4, 4, 4, 4, 0.
The date is 2023-11-13. Monday. Today's weather is sunny, and air quality is good. From 07:00 to 09:00 traffic is busy, from 16:00 to 18:00 traffic is busy. From 12:00 to 13:00 the user feel fatigued, at 17:00 the user feel fatigued, from 19:00 to 21:00 the user feel fatigued. The user has planned a date night. Starting from 00:00 on date 2023-11-13, the user's hourly trajectory is: 0, 0, 0, 0, 1, 1, 1, 0, 3, 2, 2, 2, 2, 2, 2, 2, 2.
Objective:
Determine if the user is likely to visit location 3 in the next 5 hrs?
Answer 1 if answer is Yes. Otherwise, answer 0. |
| Answer | [1] |

Table 27: POI prediction

| | | |
|---|---|---|
| Example | Help me answer question regarding a user's trajectory: 
 Given Information: 
 The coordinates of locations are as follows: Location 0 coordiante: (34.051645489083, -118.243997724533). 
 Location 1 coordiante: (34.018319580967, -118.492826418854). 
 Location 2 coordiante: (34.050016, -118.261176). 
 Location 3 coordiante: (34.033664, -118.229166). 
 Location 4 coordiante: (34.072917, -118.35729). 
 Location 5 coordiante: (34.10156947286, -118.340960963788). 
 Location 6 coordiante: (33.99535, -118.475103). 
 Location 7 coordiante: (34.132390527116, -118.280364162419). 
 Location 8 coordiante: (34.043871172843, -118.266447633989). 
 The user's trajectory and information of each day are as follows: 
 The date is 2023-11-11. Saturday. Today's weather is sunny, and air quality is good. From 12:00 to 18:00 traffic is busy. At 13:00 the user feel fatigued, at 15:00 the user feel fatigued, at 17:00 the user feel fatigued, from 19:00 to 21:00 the user feel fatigued. The user has planned a date night. Starting from 22:00 on date 2023-11-11, the user's hourly trajectory is: 5, 0. 
 The date is 2023-11-12. Sunday. Today's weather is cloudy, and air quality is good. From 12:00 to 18:00 traffic is busy. From 19:00 to 21:00 the user feel fatigued. Today there is a big sales event, like Black Friday. Starting from 00:00 on date 2023-11-12, the user's hourly trajectory is: 0, 0, 0, 0, 0, 0, 6, 6, 0, 4, 4, 4, 4, 3, 4, 4, 4, 4, 4, 4, 4, 4, 4, 0. 
 ... ... 
 The date is 2023-11-17. Friday. Today's weather is cloudy, and air quality is good. From 07:00 to 09:00 traffic is busy, from 16:00 to 18:00 traffic is busy. At 13:00 the user feel fatigued, from 19:00 to 21:00 the user feel fatigued. A popular movie premieres today. Starting from 00:00 on date 2023-11-17, the user's hourly trajectory is: 0, 0, 0, 0, 0, 0, 2, 2, 2, 2, 2, 2, 2, 2, 2, 2, 0, 0, 5, 0, 0, 5, 0. 
 The date is 2023-11-18. Saturday. Today's weather is sunny, and air quality is good. From 12:00 to 18:00 traffic is busy. At 15:00 the user feel fatigued, from 19:00 to 21:00 the user feel fatigued. Today there is a big sales event, like Black Friday. Starting from 00:00 on date 2023-11-18, the user's hourly trajectory is: 0, 0, 0, 0, 0, 0, 4, 4, 4, 4, 0, 0, 0, 4, 4, 4, 4, 4, 4, 4, 4, 4, 4. 
 Objective: 
 Determine if the user is likely to go shopping in the next 5 hrs? 
 Answer 1 if answer is Yes. Otherwise, answer 0. |
| Answer | [1] |

Table 28: Intent prediction

| model | gpt-4o | gpt-4.1 | gpt-4.5-preview | gpt-4o-mini | Llama-4 | Llama-3-8b | Mistral-7B | deepseek-chat | o3-mini | o3 |
|---|---|---|---|---|---|---|---|---|---|---|
| L-L-contains | **0.000** | **0.000** | **0.000** | **0.000** | **0.000** | 0.500 | *0.458* | 0.042 | **0.000** | **0.000** |
| L-L-crosses | 0.208 | **0.000** | *0.125* | 0.375 | 0.500 | 0.316 | 0.625 | 0.417 | 0.083 | 0.250 |
| L-L-equals | 0.043 | **0.000** | **0.000** | *0.083* | **0.000** | 0.542 | 0.500 | **0.000** | **0.000** | **0.000** |
| L-L-intersects | 0.208 | *0.125* | 0.042 | *0.125* | 0.304 | 0.667 | 0.458 | 0.208 | **0.000** | **0.000** |
| L-L-overlaps | 0.042 | **0.000** | **0.000** | 0.292 | *0.130* | 0.500 | 0.500 | 0.042 | **0.000** | **0.000** |
| L-Pt-intersects | **0.000** | *0.375* | **0.000** | 0.208 | **0.000** | *0.375* | 0.708 | **0.000** | **0.000** | **0.000** |
| L-Pg-crosses | 0.167 | 0.208 | *0.083* | 0.167 | 0.304 | 0.625 | 0.333 | 0.167 | 0.042 | **0.000** |
| L-Pg-intersects | 0.136 | 0.042 | 0.042 | 0.292 | 0.208 | 0.375 | 0.375 | *0.083* | *0.083* | **0.000** |
| L-Pg-within | 0.043 | **0.000** | **0.000** | *0.167* | 0.261 | 0.542 | 0.292 | 0.208 | **0.000** | **0.000** |

| model | gpt-4o | gpt-4.1 | gpt-4.5-preview | gpt-4o-mini | Llama-4 | Llama-3-8b | Mistral-7B | deepseek-chat | o3-mini | o3 |
|---|---|---|---|---|---|---|---|---|---|---|
| During | 0.167 | 0.167 | **0.000** | 0.167 | 0.179 | 0.655 | 0.300 | *0.067* | **0.000** | 0.033 |
| Finishes | **0.000** | **0.000** | **0.000** | *0.300* | 0.034 | 0.552 | 0.333 | **0.000** | **0.000** | **0.000** |
| Is-equal-to | **0.000** | **0.000** | *0.034* | **0.000** | 0.067 | 0.414 | 0.433 | 0.033 | **0.000** | **0.000** |
| Meets | 0.100 | 0.033 | *0.067* | 0.133 | 0.310 | 0.414 | 0.633 | 0.233 | **0.000** | **0.000** |
| Overlaps-with | 0.429 | 0.167 | 0.067 | 0.500 | 0.393 | 0.621 | 0.467 | *0.333* | 0.067 | 0.067 |
| Precedes | *0.034* | 0.033 | **0.000** | 0.167 | **0.000** | 0.400 | 0.433 | **0.000** | **0.000** | **0.000** |
| Starts | **0.000** | **0.000** | **0.000** | 0.033 | 0.200 | 0.433 | 0.700 | *0.167* | **0.000** | 0.033 |

Table 29: Spatiotemporal relationship reasoning. First table: results grouped by spatial relationship (L: Linestring, Pt: Point, Pg: Polygon). Second table: results grouped by temporal relationship.

| model | gpt-4o | gpt-4.1 | gpt-4.5-preview | gpt-4o-mini | Llama-4 | Llama-3-8b | Mistral-7B | deepseek-chat | o3-mini | o3 |
|---|---|---|---|---|---|---|---|---|---|---|
| Contains | *0.108* | 0.027 | *0.108* | 0.135 | **0.000** | *0.108* | 0.568 | 0.027 | **0.000** | **0.000** |
| Crosses | **0.000** | **0.000** | **0.000** | 0.087 | **0.000** | 0.478 | *0.391* | **0.000** | **0.000** | **0.000** |
| Equals | **0.000** | **0.000** | **0.000** | **0.000** | **0.000** | **0.000** | **0.000** | **0.000** | **0.000** | **0.000** |
| Intersects | **0.000** | **0.000** | **0.000** | **0.000** | **0.000** | *0.500* | 0.155 | **0.000** | **0.000** | **0.000** |
| Overlaps | 0.375 | 0.312 | **0.000** | 0.500 | *0.071* | 0.438 | 0.062 | 0.312 | **0.000** | **0.000** |
| Touches | 0.017 | **0.000** | **0.000** | *0.034* | **0.000** | 0.483 | 0.207 | *0.034* | **0.000** | **0.000** |
| Within | 0.135 | *0.081* | 0.027 | 0.297 | **0.000** | 0.405 | 0.405 | **0.000** | **0.000** | **0.000** |

Table 30: Spatial relationship reasoning by predicates.

| model | gpt-4o | gpt-4.1 | gpt-4.5-preview | gpt-4o-mini | Llama-4 | Llama-3-8b | Mistral-7B | deepseek-chat | o3-mini | o3 |
|---|---|---|---|---|---|---|---|---|---|---|
| Precedes | **0.000** | **0.000** | **0.000** | **0.000** | **0.000** | *0.667* | 0.333 | **0.000** | **0.000** | **0.000** |
| Is-preceded-by | 0.556 | *0.222* | **0.000** | 0.778 | 0.111 | 0.667 | 0.333 | 0.444 | **0.000** | **0.000** |
| Meets | **0.000** | **0.000** | **0.000** | **0.000** | **0.000** | 0.333 | *0.667* | **0.000** | **0.000** | **0.000** |
| Is-met-by | **0.000** | **0.000** | **0.000** | **0.000** | **0.000** | *0.556* | 0.111 | **0.000** | **0.000** | **0.000** |
| Overlaps-with | **0.000** | **0.000** | **0.000** | **0.000** | **0.000** | *0.556* | 0.222 | **0.000** | **0.000** | **0.000** |
| Is-overlapped-by | *0.333* | **0.000** | **0.000** | 0.222 | 0.556 | 0.444 | 0.556 | **0.000** | **0.000** | **0.000** |
| Starts | **0.000** | **0.000** | **0.000** | **0.000** | **0.000** | *0.556* | 0.333 | **0.000** | **0.000** | **0.000** |
| Is-started-by | **0.000** | **0.000** | **0.000** | **0.000** | **0.000** | *0.667* | 0.333 | **0.000** | **0.000** | **0.000** |
| During | **0.000** | **0.000** | **0.000** | **0.000** | **0.000** | 0.333 | **0.000** | **0.000** | **0.000** | **0.000** |
| Contains | **0.000** | **0.000** | **0.000** | **0.000** | **0.000** | 0.222 | *0.556* | **0.000** | **0.000** | **0.000** |
| Finishes | **0.000** | **0.000** | **0.000** | **0.000** | **0.000** | *0.667* | 0.333 | **0.000** | **0.000** | **0.000** |
| Finished-by | **0.000** | **0.000** | **0.000** | **0.000** | **0.000** | *0.778* | 0.222 | **0.000** | **0.000** | **0.000** |
| Is-equal-to | **0.000** | **0.000** | **0.000** | **0.000** | **0.000** | **0.000** | **0.000** | **0.000** | **0.000** | **0.000** |

Table 31: Temporal relationship reasoning by Allen's temporal algebra.

