# OpenReview forum: "Benchmarking Spatiotemporal Reasoning in LLMs and Reasoning Models: Capabilities and Challenges"
_NeurIPS.cc/2025/Datasets_and_Benchmarks_Track — NeurIPS 2025 Datasets and Benchmarks Track poster_

### Official Review · Reviewer_mhtH · 2025-06-05

**Rating:** 5
**Confidence:** 3

**Summary:**

The paper introduces STARK, a benchmark that measures spatio-temporal reasoning in large language models (LLMs) and large reasoning models (LRMs). STARK evaluates LLMs across three levels of reasoning complexity: state estimation, spatiotemporal reasoning, and world-knowledge-aware reasoning. STARK contains 26 tasks / 14 552 instances from diversive sensors. Results show that (1) LRMs show robust performance across tasks with different complexity levels. (2) For world-knowledge-aware reasoning, the gap between LRMs and LLMs narrows.

**Dataset Code Accessibility:**

Yes

**Ethical Considerations:**

No, there are no or only very minor ethics concerns

**Final Justification:**

I appreciate the authors' detailed response to my questions. I am satisfied with the response.

**Limitations Weaknesses:**

1. 1315/14552 questions were actually executed “due to financial constraints” (lines 176-179). This may under-sample several Tier 3 tasks whose absolute counts are small (e.g., ETA has only a handful of items), so error bars are large.

2.  The LRMs evaluated are all closed models. Reproducibility of the headline results therefore depends on commercial APIs. The authors can include an open-source reasoning model for longevity.

3. First-principle baselines are strong for Tier 1 but missing for many Tier 2/3 tasks, leaving random-guess as the only comparator (section 3.3). It could be hard to gauge real difficulty.

4. There are some minor issues in presentation. In Fig. 1, the labels overlap in the pie chart.

**Strengths Contributions:**

1. Existing suites focus on either spatial or temporal cues or provide only multiple-choice QA. STARK unifies space, time and world-knowledge, and requires open-ended numeric answers or tool use.

2. STARK introduces several novel task setups. The authors design fine-grained three-tier hierarchy for evaluation. The benchmark covers heterogeneous sensor modalities with noise injection. STARK is the first to combine ArcGIS DE-9IM + Allen algebra in a single benchmark.

3. Authors provide detailed analysis based on 3 LRMs and 8 LLMs. For example, they showcase how to diagnose GDOP failure cases for multilateration (Fig. 6) and EKF sensitivity (Fig. 7).

4. The writing is concise and well-structured. Figures are informative.

---

> ### Author Rebuttal · Authors · 2025-07-31
>
> >**Q1**: 1315/14552 questions were actually executed “due to financial constraints” (lines 176-179). This may under-sample several Tier 3 tasks whose absolute counts are small (e.g., ETA has only a handful of items), so error bars are large.
>
> **Response**: We appreciate the reviewer’s concern regarding potential undersampling in early evaluations. To address this, and facilitated by the significant lowering by OpenAI of the cost of the o3 model, we have completed **full-benchmark** evaluations for all models with one exception. The exception is gpt-4.5 model which OpenAI has deprecated and removed access, preventing us from running the full benchmark on it. The updated results show that model performance on the full-scale benchmark is consistent with the trends observed in the small-scale evaluation (Table 3, line 197).
>
> |model|gpt-4o|gpt-4.1|gpt-4o-mini|Llama-4|deepseek|Llama-3-8b|Mistral-7B|o3-mini|o4-mini|o3|
> |-|-|-|-|-|-|-|-|-|-|-|
> |Tier1|||||
> |S-impute|0.117|0.008|0.008|0.893|0.003|0.013|0.842|`0.001`|*0.001*|**0.000**|
> |ST-forecast|*0.147*|`0.147`|0.164|0.166|**0.141**|0.176|0.199|0.158|0.161|0.173|
> |ST-impute|`0.101`|0.104|0.127|0.108|0.105|0.143|0.173|*0.102*|**0.101**|0.116|
> |T-impute|0.073|`0.069`|0.079|**0.067**|0.076|0.082|0.132|0.070|*0.070*|0.087|
> |Loc-range|2.177|1.562|2.812|0.493|2.080|2.962|2.743|*0.056*|`0.044`|**0.043**|
> |Loc-bearing|2.895|2.620|2.926|1.688|2.223|3.032|2.813|*0.166*|**0.151**|`0.153`|
> |Loc-range-bearing|1.855|1.645|1.932|0.116|0.119|3.031|2.851|`0.062`|*0.063*|**0.053**|
> |Loc-proximity|1.065|1.034|1.054|1.034|0.958|1.413|1.801|*0.956*|`0.878`|**0.858**|
> |T-loc-event|1.334|0.638|0.833|0.398|0.389|1.171|2.229|`0.153`|*0.191*|**0.064**|
> |S-loc-event|2.790|2.584|2.890|2.746|2.772|2.924|2.765|`0.292`|*0.593*|**0.184**|
> |Track-range|2.891|2.934|3.610|2.141|2.642|4.099|3.789|`0.656`|*0.852*|**0.556**|
> |Track-bearing|3.644|3.410|3.628|2.687|2.090|4.272|3.611|`0.822`|*0.866*|**0.791**|
> |Track-range-bearing|1.179|2.048|1.357|0.836|0.138|4.151|3.820|*0.128*|`0.124`|**0.119**|
> |Track-proximity|**2.020**|`2.024`|2.097|2.209|2.156|2.657|2.763|2.276|2.205|*2.083*|
> |S-track-event|3.340|3.750|3.619|3.493|3.647|4.040|3.488|`1.265`|*1.815*|**0.931**|
> |T-track-event|0.329|*0.136*|0.255|0.447|0.141|0.383|0.428|`0.114`|0.138|**0.080**|
> |Tier2|||||
> |S relationship|0.074|*0.053*|0.169|0.064|0.069|0.334|0.311|**0.000**|`0.009`|**0.000**|
> |T relationship|**0.000**|**0.000**|`0.021`|**0.000**|**0.000**|0.467|*0.363*|**0.000**|**0.000**|**0.000**|
> |S Trelationship|0.133|*0.042*|0.187|0.214|0.130|0.434|0.507|**0.026**|`0.029`|0.057|
> |Tier3|||||
> |Landmark direction|0.259|0.383|0.296|0.400|0.321|0.481|0.617|*0.150*|`0.074`|**0.037**|
> |Intent prediction|`0.207`|0.332|0.398|0.253|0.237|0.506|0.361|0.282|*0.224*|**0.195**|
> |Landmark proximity|*0.282*|`0.266`|0.577|0.552|0.394|0.419|0.481|0.307|**0.216**|0.303|
> |POI prediction|0.162|*0.158*|0.465|`0.154`|0.249|0.423|0.527|0.212|0.199|**0.124**|
> |Route planning|**0.000**|**0.000**|`0.136`|**0.000**|**0.000**|**0.000**|0.333|*0.148*|**0.000**|**0.000**|
> |ETA|`0.020`|*0.030*|0.139|0.099|`0.020`|0.238|0.475|*0.030*|**0.000**|**0.000**|
> |Route segment dur.|0.136|*0.099*|0.148|0.148|`0.086`|0.531|0.593|0.173|0.185|**0.074**|
>
> - To accommodate different research needs and computational budgets, we plan to publicly release the two versions of the benchmark:
> 	- STARK-S: a lightweight subset (1,315 samples) for rapid experimentation and low-cost evaluation;
> 	- STARK-L: the full benchmark (14,552 samples) for comprehensive analysis.
> - Additionally, we further include a detailed evaluation budget breakdown per model on STARK-L to support reproducibility and assist future users in estimating evaluation costs.
>
> | Model         | Cost per run | Provider     |
> |--|-|-|
> | O3*| $759.8| OpenAI|
> | O3-mini| $442.9| OpenAI|
> | O4-mini| $352.6| OpenAI|
> | GPT-4.1| $237.5| OpenAI|
> | GPT-4o| $284.1| OpenAI|
> | GPT-4o-mini| $18.3| OpenAI|
> | LlaMA-4| $44.4| Together.ai  |
> | LlaMA-3-8B| $9.21| Together.ai  |
> | Mistral-7| $25.1| Together.ai  |
> *The cost of o3 is reduced by 80% since 06/2025.
> >**Q2**:  The LRMs evaluated are all closed models. Reproducibility of the headline results therefore depends on commercial APIs. The authors can include an open-source reasoning model for longevity.
>
> **Response**: We agree that relying solely on closed-source LRMs may raise concerns about long-term reproducibility. To address this, we conducted additional experiments using open-source reasoning models, including R1-Distill-Qwen-1.5B, R1-Distill-Qwen-14B, R1-Distill-Llama-70B, and Llama-3.3-70B (base model of R1-Distill-Llama-70B) [1].
>
> |model|R1-Distill-Qwen-1.5B|R1-Distill-Qwen-14B|R1-Distill-Llama-70B|Llama-3.3-70B|
> |---|---|---|---|---|
> |Tier1|||||
> |S-impute|0.483|**0.006**|*0.014*|`0.011`|
> |ST-forecast|0.329|`0.204`|**0.168**|*0.269*|
> |ST-impute|0.339|`0.131`|**0.115**|*0.177*|
> |T-impute|0.187|**0.069**|`0.088`|*0.169*|
> |Loc-range|*3.032*|`0.695`|**0.646**|3.166|
> |Loc-bearing|*3.130*|**0.486**|`0.811`|3.389|
> |Loc-range-bearing|3.125|`0.233`|**0.191**|*2.605*|
> |Loc-proximity|3.261|`1.449`|**1.134**|*2.191*|
> |T-loc-event|2.912|**0.469**|`0.469`|*0.749*|
> |S-loc-event|3.709|**1.723**|`2.138`|*3.031*|
> |Track-range|3.826|`3.272`|**2.654**|*3.518*|
> |Track-bearing|*3.826*|**3.092**|`3.193`|3.849|
> |Track-range-bearing|3.826|`2.635`|**2.059**|*3.284*|
> |Track-proximity|3.742|*2.915*|**2.461**|`2.806`|
> |S-track-event|*3.826*|*3.826*|**3.267**|`3.561`|
> |T-track-event|1.000|*0.969*|**0.404**|`0.722`|
> |Tier2|||||
> |S relationship|0.359|**0.072**|`0.074`|*0.128*|
> |T relationship|0.735|`0.034`|**0.000**|*0.068*|
> |ST relationship|0.556|**0.147**|`0.167`|*0.183*|
> |Tier3|||||
> |Landmark direction|*0.680*|`0.280`|**0.259**|`0.280`|
> |Intent prediction|`0.480`|*0.560*|**0.280**|**0.280**|
> |Landmark proximity|*0.640*|`0.560`|**0.400**|0.720|
> |POI prediction|0.520|*0.280*|`0.240`|**0.200**|
> |Route planning|*0.760*|`0.080`|`0.080`|**0.040**|
> |ETA|0.720|*0.120*|**0.000**|`0.040`|
> |Route segment dur.|1.000|**0.200**|`0.240`|*0.280*|
>
> These open-source models serve two purposes:
> 	- Reproducibility: They provide the community with accessible references that can be freely tested, audited, and extended.
> 	- New observations: They allow controlled comparison between LLMs and LRMs of similar size, architecture, and training origin.
>
> Our findings show:
>
> - The claim in the submission regarding the positive correlation between model size and performance is further supported by the new data. In particular, for Tier-3 world-knowledge reasoning tasks, we observe a substantial decrease in error rates as the model size increases from 1.5B to 70B parameters.
>
> - Furthermore, a direct comparison between R1-Llama-70B LRM and Llama-3.3-70B (the base model of R1-Llama-70B) highlights the benefits of reasoning alignment [1]. For instance, the R1-Llama-70B model:
> 	- Reduces the error on range-based location tasks by 75%, and
> 	- Matches or exceeds Llama-3.3-70B’s performance on world-knowledge reasoning tasks.
>
> These results support our main findings in the draft and also provide a reproducible reference for the community.
>
> >**Q3**:  First-principle baselines are strong for Tier 1 but missing for many Tier 2/3 tasks, leaving random-guess as the only comparator (section 3.3). It could be hard to gauge real difficulty.
>
> **Response**: We appreciate the reviewer’s concern regarding the availability of strong baselines for Tier 2 and Tier 3 tasks. We would like to clarify the following:
> - Tier 2 (Spatiotemporal reasoning over states):
> Many of these tasks are solvable in polynomial time with deterministic algorithms. For example, spatial reasoning problems (e.g., point-in-polygon, intersection, containment) can be solved using the `shapely` library, which serves as a near-perfect oracle. These tools effectively establish a 100% accuracy baseline, offering a clear upper bound for task difficulty. Therefore, we did not use these tools as the baselines because of the unfair comparison with LLMs/LRMs.
> - Tier 3 (World-knowledge-aware reasoning):
> Similarly, for spatial queries involving landmarks (e.g., nearest landmark, directional relation), GIS services like ArcGIS can provide precise answers. However, for tasks like POI prediction and intent prediction, where the tools are not applicable, we implemented a probabilistic baseline using a finite state machine (FSM) [2]. This model encodes general behavior transitions **without** access to contextual cues or external knowledge.
>
> The following table shows the FSM baseline error compared to model performance:
>
> | Task              | FSM | o3 | GPT-4o |
> |------------------|-------------|------------|----------------|
> | POI Prediction   | 0.1667      | 0.0000     | 0.0000         |
> | Intent Prediction| 0.2916      | 0.0000     | 0.2000         |
>
> These results underscore that while FSM offers a non-trivial and explainable baseline, it lacks the contextual awareness necessary for accurate reasoning, where models like o3 and GPT-4o demonstrate superiority.
>
> We will clarify in the revised manuscript that many Tier 2/3 tasks do have strong symbolic or tool-based baselines (beyond random guess) to eliminate any confusion.
>
> >**Q4**:  There are some minor issues in presentation. In Fig. 1, the labels overlap in the pie chart.
>
> **Response**: Thank you for pointing this out. We will revise Figure 1 to resolve the label overlap issue in the pie chart and ensure improved readability in the final version.
>
> [1] Guo, Daya, et al. "Deepseek-r1: Incentivizing reasoning capability in llms via reinforcement learning." arXiv preprint arXiv:2501.12948 (2025).
>
> [2] Fernández-Caballero, Antonio, José Carlos Castillo, and José María Rodríguez-Sánchez. "Human activity monitoring by local and global finite state machines." Expert Systems with Applications 39.8 (2012): 6982-6993.

---

> > ### Comment · Reviewer_mhtH · 2025-08-02
> >
> > I appreciate the authors' detailed response to my questions. I am satisfied with the response.

---

### Official Review · Reviewer_waoq · 2025-06-14

**Rating:** 5
**Confidence:** 4

**Summary:**

This paper proposes STARK, a pioneering benchmark for evaluating spatiotemporal reasoning in large language models (LLMs) and large reasoning models (LRMs). Through a hierarchical task design, it systematically assesses model capabilities across the entire perception-to-decision pipeline. Compared to existing benchmarks such as GeoQA, STARK offers substantial advances in task diversity, evaluation realism, and experimental depth, making it highly relevant for CPS and agent-based AI applications.
The experiments are thorough, the insights are significant, and the benchmark resources are fully open-sourced. However, there are areas that require refinement, including attribution of model capabilities, methodological clarity, and consistent terminology usage. Overall, this work represents a strong and timely contribution to the field and is recommended for acceptance after minor revisions.

**Dataset Code Accessibility:**

Yes

**Dataset Code Comments:**

The data set can be obtained

**Ethical Considerations:**

No, there are no or only very minor ethics concerns

**Final Justification:**

Thanks for providing the response. They are clear now.

**Limitations Weaknesses:**

1 Terminological Confusion and Presentation Issues

The paper inconsistently uses the terms “DA (Direct Answering)” and “direct reasoning” without clearly distinguishing their meanings. This creates ambiguity between answer formats and task reasoning levels. It is recommended to unify terminology and explicitly define these concepts in the methodology section.

2 Insufficient Attribution of Model Capability

The paper speculates that the superior performance of LRM models is due to larger parameter size , but does not provide comparisons of architecture or training data between o3 and o3-mini, weakening the rigor of its claims.

Differences between LRM and LLM architectures are only referenced via system cards , without explaining whether specialized geometric modules or architecture choices contributed to performance gains. Further architectural analysis or ablation studies are recommended.

3 Experimental Coverage and Generalization Limitations

Due to cost constraints, only 1,315 out of 14,552 tasks are evaluated, which may limit the statistical significance of findings;

Tier-3 tasks such as POI prediction rely on synthetic trajectories generated by a finite state machine , without validation on real user behavior data. This may overestimate real-world generalization.

4 Lack of Methodological Detail and Error Analysis

The training pipeline for LRM models is not described. It remains unclear how temporal and spatial signals are encoded, or whether specialized mechanisms are used for spatiotemporal input processing;

Failure cases in CI mode (especially in tracking tasks, Figure 8) are not systematically analyzed. A taxonomy of errors and failure modes would better inform future improvements.

**Strengths Contributions:**

1 Pioneering Benchmark Design (STARK)

Hierarchical Task Structure: STARK innovatively divides spatiotemporal reasoning into three tiers of complexity—state estimation → reasoning over states → knowledge-enhanced reasoning—providing full-stack evaluation from low-level perception to high-level decision-making . In contrast to prior benchmarks like GeoQA which focus on a single dimension, STARK integrates geometric reasoning, five sensor modalities, and real-world knowledge in a unified framework.

Multi-modal Sensor Fusion: The benchmark includes five sensor types (range, bearing, event, proximity, status), covering a broad range of realistic perception scenarios.

2 Realistic and Practical Evaluation Setup

Open-ended Answer Format: All tasks require models to produce direct answers or code outputs, rather than selecting from multiple-choice options . This design better reflects real-world CPS demands. The scale of 14,552 task instances enhances statistical reliability .

Tool-usage Evaluation (Code Interpreter): This is the first benchmark to systematically compare direct answering (DA) and code interpreter (CI) modes . CI significantly improves localization performance but degrades tracking accuracy , revealing critical challenges in LLM-toolchain integration.

3 Rigorous Experimental Analysis

Sensor data is simulated on a 10×10 grid with realistic noise patterns , including range, bearing, and event-based inputs—supporting generalizability of conclusions;

Comparative evaluation shows LRM models outperform classical optimization in localization, and surpass EKF in tracking due to superior noise robustness ;

Prompts, templates, and evaluation setups are clearly documented in the appendix, ensuring reproducibility;

4 Valuable Resource Contribution

Full Open-Source Release: The paper provides complete access to dataset generation code and evaluation scripts , supporting reproducibility and facilitating future research efforts.

Transparent Cost Reporting: Detailed API usage costs are disclosed , offering practical reference for future benchmark design.

---

> ### Author Rebuttal · Authors · 2025-07-31
>
> > **Q1**: The paper inconsistently uses the terms “DA (Direct Answering)” and “direct reasoning”. It is recommended to unify terminology and explicitly define these concepts.
>
> **Response**: Thank you for pointing out the terminological inconsistency. We propose to change the phrase “direct reasoning” in Line 60, which is the only place the phrase is used, to “spatiotemporal reasoning” to avoid confusion with the term “Direct Answering.”
>
> >**Q2**: (1) The paper attributes LRM performance to model size but lacks comparisons of architecture or training data between o3 and o3-mini, weakening the claim. (2) Architectural differences are only mentioned via system cards, without analyzing the impact of specialized modules. Deeper analysis or ablations are recommended.
>
> **Response**: Thank you for raising this important point regarding the attribution of LRM performance. We note that the commercial LRMs (o3 and o3-mini) are closed-source, accessible only via API. As a result, we are unable to isolate which architectural or data-driven factors contribute to the observed performance differences. To address this limitation, we have conducted additional experiments on open-source LRM variants:
>
> - **Empirical Comparison Across Model Sizes**: To strengthen our claim about the relationship between model size and performance, we conducted additional experiments on open-source LRMs and their LLM counterparts, including R1-Distill-Qwen-1.5B, R1-Distill-Qwen-14B, R1-Distill-Llama-70B, and Llama-3.3-70B (the base model of R1-Distill-Llama-70B [1]). The results show a consistent performance gain for LRMs as model size increases. Notably, on tier-3 world-knowledge reasoning tasks, the error rate significantly decreases. Due to space constraints, we refer the reviewer to the table provided in our response to **Q2** of `Reviewer mhtH`.
>
> - **The Impact of Reasoning Alignment [1]**: After comparing R1-Distill-Llama-70B with its base model Llama-3.3-70B, we observed that LRMs fine-tuned via reasoning alignment consistently outperform their base LLM counterparts. These results suggest that the performance gain is likely due to the reasoning alignment [1], given the same model size and architecture, and pretrained data.
>
> - **Architecture Insights**: We agree that clearer attribution of performance to architectural or training data differences would improve rigor. We will expand the discussion to better discuss model architecture and training paradigms, given the new experiments and observations:
> 	- Model size plays an important role. As confirmed by the comparison between the 1.5B, 14B, and 70B models, the ability to perform world-knowledge reasoning increases.
> 	- Reasoning alignment matters. LRMs outperform LLMs of the same size, especially on complex tier 1 and 2 spatiotemporal tasks.
>
> > **Q3**: Due to cost constraints, only 1,315 out of 14,552 tasks are evaluated, which may limit the statistical significance of findings;
>
> **Response**: Thank you for highlighting these important considerations regarding experimental stability and generalization.
> - **Full Benchmark Evaluation**: While at the time we were constrained due to the high cost of the o3 and gpt-4.5 models, fortunately since then, OpenAI has significantly reduced the cost of o3 and also deprecated the gpt-4.5 model (which is no longer available). We have therefore evaluated the full benchmark (all the 14,552) tasks, but had to exclude gpt-4.5. Our updated results show that model performance on the small-scale subset reported in the submission (Table 3, Line 197) is generally consistent with performance on the full-scale dataset, supporting the statistical validity of our earlier findings. We also provide the full evaluation budget per model. Furthermore, to support a wider range of research scenarios, we plan to release both STARK-S (1.3k samples, rapid evaluation) and STARK-L (14k samples, comprehensive evaluation), enabling the community to choose the benchmark size that fits their cost budget. Due to space limits, please see the full benchmark evaluation and budget tables in our response to **Q1** of `Reviewer mhtH`.
>
> >**Q4**:  Tier-3 tasks such as POI prediction rely on synthetic trajectories generated by a finite state machine. This may overestimate real-world generalization.
>
> **Response**: Thank you for raising this important point. While we agree that the use of finite state machines (FSMs) to model the movement intention of the mobile entity could limit real-world generalization, this approach offers control and transparency [4, 5], allowing us to model intent-driven transitions, context-aware behaviors, and mobility patterns essential for evaluating high-level, world-knowledge-based reasoning. While human-subject studies could provide annotations of intent and social context, such human-in-the-loop validation falls outside the scope of this paper and will introduce its own set of challenges (cost, privacy, uncertainty) without adding commensurate value.
>
> Additionally, existing real-world datasets of mobility traces, such as T-Drive [2] and GeoLife [3], primarily include trajectories with timestamps but lack the rich contextual signals, like traffic conditions, weather, or social events, required for our reasoning tasks. As such, they are not directly suitable for STARK. Nonetheless, we agree that real-world validation is an important next step and we will highlight it as a promising direction for future work to assess model generalization and robustness.
>
> >**Q5**: The training pipeline for LRM models is not described. It remains unclear how temporal and spatial signals are encoded, or whether specialized mechanisms are used;
>
> **Response**: Thank you for raising this important point. We acknowledge that understanding how LRM models encode spatial and temporal signals is highly relevant. However, as many of the LRM models are proprietary and lack a detailed specification of their pipeline or transparency about the training data, it is difficult to authoritatively state how their training pipeline is handled. Obtaining this information would require significant reverse engineering and analysis of these black-box model architectures, training pipelines, and data. Besides being tangential to the primary objective of this paper, it is unclear whether such reverse engineering is practically feasible in a reliable manner. Instead, our goal is to evaluate the spatiotemporal reasoning capabilities of state-of-the-art models, both proprietary and open-source,  through a structured benchmark. That said, we agree that a deeper understanding of the effects of the underlying training and encoding mechanisms could offer additional valuable insights into model behaviors on spatiotemporal reasoning tasks. We will include a discussion note to highlight this as a potential direction, should the paper be accepted.
>
> >**Q6**: Failure cases in CI mode (especially in tracking tasks, Figure 8) are not systematically analyzed. A taxonomy of errors and failure modes would better inform future improvements.
>
> **Response**: We thank the reviewer for the suggestion to provide a taxonomy of failure modes. We analyzed CI-mode failures for o3-mini, focusing on tracking tasks with large errors. Building on [6], we identify six key failure types, categorized as root failures (primary cause) and side failures (secondary cause or consequences of the root error):
> 1. Wrong Problem Formulation: Misunderstanding of the task framing.
> 2. Numerical / Optimization Issues: Failures due to non-convergence, local minima, or instability during optimization.
> 3. Data-Handling Bugs: Mistakes in indexing, type conversion, or NaN handling during input preprocessing.
> 4. Protocol Mismatches: Violations of required output formatting (e.g., messing with Python coding format).
> 5. Lack of Sanity / Physics Checks: Implausible or physically impossible results generation.
> 6. Ill-Conditioned Geometry: Problem setups with poor sensor placement or ambiguous configurations that amplify numerical errors.
>
> |**Task**|**Instance#**|**RMSE**|**RootFailure**|**SideFailure(s)**|
> |-|-|-|-|-|
> |Spatialtracking-event|7|7847.5|2|5,1|
> |Spatialtracking-event|17|5.8|2|5,1|
> |Temporaltracking-event|3|131.3|2|5|
> |Temporaltracking-event|4|102.9|5|2,4|
> |Temporaltracking-event|17|68.4|5|2,1,4|
> |Tracking-range|2|5.2|3|5|
> |Tracking-range|9|5.0|6|5,2|
>
> Insights and Recommendations:
> 1. Numerical / Optimization issues (Type 2) are the most frequent root cause of failure. These often manifest as divergence or incorrect local minima in multilateration/triangulation problems.
> 2. Lack of sanity checks (Type 5) frequently appears as a side failure. For example, missing bounds or regularization often lets the optimizer return implausible values (e.g., speed > 3000 m/s).
> 3. Simple additions such as physical limits and result checks could prevent many of these failures.
>
> While we manually inspect failure cases, we anticipate using additional tools to scale this analysis to more tasks in the future through LLM-aided error classifiers and rule-based evaluators.
>
> [1] Guo, Daya, et al. "Deepseek-R1: Incentivizing reasoning capability in LLMs via reinforcement learning." arXiv preprint arXiv:2501.12948 (2025).
>
> [2] Yuan, Jing, et al. "Driving with knowledge from the physical world." Proceedings of the 17th ACM SIGKDD. 2011.
>
> [3] Zheng, Yu, et al. "Mining interesting locations and travel sequences from GPS trajectories." Proceedings of the 18th WWW. 2009.
>
> [4] Mohmed, et al." Enhanced fuzzy finite state machine for human activity modelling and recognition." Journal of Ambient Intelligence and Humanized Computing 11.12 (2020): 6077-6091.
>
> [5] Fernández-Caballero, et al."Human activity monitoring by local and global finite state machines." Expert Systems with Applications 39.8 (2012): 6982-6993.
>
> [6] Wang, Zhijie, et al. "Where do large language models fail when generating code?." arXiv e-prints (2024): arXiv-2406.

---

> > ### Author Response · Authors · 2025-08-06
> > **Follow‑up on discussion**
> >
> > Dear Reviewer,
> >
> > We sincerely thank you for your feedback and suggestions. We have carefully addressed each of your points in our response, incorporating additional experiments, clarifications, and discussion.
> >
> > Could you kindly let us know if we have addressed all your concerns and whether you are satisfied with our response?
> >
> > Thanks,
> >
> > Paper 646 authors

---

> > > ### Comment · Reviewer_waoq · 2025-08-09
> > > **I have raised the score**
> > >
> > > Thank you for your detailed and thoughtful responses to my initial concerns. I appreciate the additional experiments (e.g., open-source LRM comparisons, full benchmark evaluation) and clarifications (e.g., terminology unification, failure mode taxonomy) provided in the rebuttal. These efforts have effectively addressed my key questions regarding model size attribution, statistical significance, real-world generalization, and error analysis. The expanded discussion on reasoning alignment and architecture insights is particularly valuable.
> > >
> > > Based on the new evidence, I have raised my score to reflect the improved rigor and completeness of the work. I look forward to seeing the updated manuscript and hope the community will benefit from the release of both STARK-S and STARK-L benchmarks.

---

> > ### Author Response · Authors · 2025-08-09
> > **Thanks for your feedback and recognition**
> >
> > Dear reviewer,
> >
> > We thank you for the constructive feedback and for recognizing our additional experiments, clarifications, and expanded discussions. We appreciate the raised score and look forward to formally releasing both STARK-S and STARK-L for the community.
> >
> > Best,
> >
> > Authors

---

### Official Review · Reviewer_oSCC · 2025-07-02

**Rating:** 5
**Confidence:** 2

**Summary:**

The paper introduces STARK, a hierarchical benchmark designed to evaluate the spatiotemporal reasoning capabilities of Large Language Models (LLMs) and Reasoning Models (LRMs) across multiple complex tasks. The evaluation demonstrates that LRMs generally outperform LLMs on localization and reasoning tasks, but larger LLMs like GPT-4.5 show competitive performance, especially when leveraging external tools like code interpreters. Moreover, the study highlights the importance of model scale and reasoning paradigms for advancing CPS applications, with all data and code openly available to facilitate reproducibility.

**Dataset Code Accessibility:**

Yes

**Ethical Considerations:**

No, there are no or only very minor ethics concerns

**Final Justification:**

I have no further questions and hope the authors optimize the manuscript accordingly.

**Limitations Weaknesses:**

While the dataset and benchmark are clearly valuable, I have a few suggestions that may improve the paper’s clarity and presentation. First, the table summarizing the results across 26 tasks is quite dense and difficult to parse, readability could be significantly improved by reorganizing or visualizing the results in a more intuitive way. For example, grouping tasks or using aggregated visual summaries might help readers quickly grasp the key takeaways. In addition, some visualizations (e.g., the use of blue shades in Figure 5) lack sufficient contrast, making it hard to distinguish between models.

Another point worth considering is the role of code interpreter (CI)-based reasoning. While the inclusion of CI is a unique and important aspect of the benchmark, the motivation and value of incorporating this mode, especially given how few existing models support it, could be more explicitly emphasized.

**Strengths Contributions:**

This paper presents a well-motivated and timely benchmark, STARK, for evaluating spatiotemporal reasoning capabilities in LLMs and LRMs. The dataset is thoughtfully designed across three reasoning levels (state estimation, temporal reasoning, and world-knowledge reasoning), covering a wide range of tasks that are highly relevant for real-world CPS applications. The authors curate an impressively large and diverse set of 14,552 challenges spanning 26 tasks and multiple sensor modalities, which is a significant effort and contribution on its own.
The findings also provide useful insights into where current models fall short, and the dataset could serve as a useful diagnostic tool in future work. Overall, this is a strong and well-executed dataset paper that fills a clear gap in the evaluation landscape.

---

> ### Author Rebuttal · Authors · 2025-07-31
>
> >**Q1**: (1) Table 3 (Line 197) is quite dense and could be improved by reorganizing or visualizing the results in a more intuitive way. (2) Some visualizations (e.g., the use of blue shades in Figure 5) lack sufficient contrast, making it hard to distinguish between models.
>
> **Response**: Thanks for the thoughtful suggestions to improve the clarity of the paper. If the paper is accepted, we will incorporate the following improvements in the camera-ready version.
>
> - **Table Readability**: We appreciate the feedback on the dense presentation of the 26-task summary table. To address this, we plan to replace the table with more intuitive visualizations, such as boxplots grouped by task categories. This will help highlight trends and performance differences across model types more effectively. The detailed numerical results will be moved to the Appendix for reference.
>
> - **Figure Contrast**: Thank you for pointing out the contrast issue in Figure 5. We will revise the color scheme to ensure better visual separation between models.
>
> Please note that as NeurIPS no longer allows including rich media, such as figures, in the author response to review comments, we are unable to provide visualizations of what we propose, but will certainly take care of these issues in the camera-ready version, should the paper be accepted.
>
> >**Q2**: Another point worth considering is the role of code interpreter (CI)-based reasoning. While the inclusion of CI is a unique and important aspect of the benchmark, the motivation and value of incorporating this mode, especially given how few existing models support it, could be more explicitly emphasized.
>
> **Response**: Thank you for highlighting the importance of clarifying the role of CI-based reasoning. We agree that the importance of CI could be better motivated and introduced. To address this, we will revise the Introduction to more explicitly articulate the motivation behind incorporating CI-based reasoning. More specifically, we will revise the Introduction to highlight how CI-based reasoning serves as a bridge between pure language models and fully autonomous tool-augmented agents. Although current support for CI is limited to a subset of models, we view it as a forward-looking capability aligned with the needs of real-world CPS applications.

---

> > ### Comment · Reviewer_oSCC · 2025-08-06
> >
> > Thanks for the response. I have no further questions and tend to maintain my rating as "Accept".

---

### Official Review · Reviewer_7AiH · 2025-07-06

**Rating:** 3
**Confidence:** 4

**Summary:**

This paper introduces STARK, a hierarchical benchmark to evaluate spatiotemporal reasoning in Large Language Models (LLMs) and Large Reasoning Models (LRMs). STARK comprises 26 tasks across three tiers—state estimation, reasoning over states, and world-knowledge-aware reasoning—incorporating diverse sensor modalities (range, bearing, event-based) and open-ended challenges (direct answering/code interpretation). Experiments show LRMs consistently outperform LLMs in geometric tasks (e.g., localization, tracking), with the LRM o3 model leading across all tasks, attributed to its larger scale. LLMs narrow the gap in knowledge-intensive tasks, with some surpassing smaller LRMs.

**Dataset Code Accessibility:**

Yes

**Ethical Considerations:**

No, there are no or only very minor ethics concerns

**Limitations Weaknesses:**

High computational costs restrict full-benchmark evaluation (e.g., o3 costs $418.5 per run).
World-knowledge tasks rely on static landmark data, lacking dynamic context (e.g., real-time traffic events).

**Strengths Contributions:**

Motivation: Addresses unmet needs in CPS applications (e.g., robot navigation, environmental monitoring) by targeting joint space-time-knowledge reasoning .
Method Rigor: Tasks balance simplicity (state estimation) with complexity (landmark-aware navigation), and CI mode evaluates tool usage .
Experimental Breadth: Evaluates 11 models across 26 tasks, revealing LRM superiority in geometric inference (e.g., o3’s 30× lower error in localization) .

---

> ### Author Rebuttal · Authors · 2025-07-31
>
> >**Q1**: High computational costs restrict full-benchmark evaluation (e.g., o3 costs $418.5 per run)
>
> **Response**: We appreciate the reviewer’s concern regarding computational cost.  We have completed full-benchmark evaluations for all models, including o3. Specifically, the query cost for o3 has decreased significantly (from `$40/M` tokens to `$8/M` tokens) since the original submission. We include the updated full-benchmark results in the revised tables (OpenAI deprecated GPT-4.5, therefore we removed it from the table). The results indicate that the performance of various models on the small-scale dataset (Table 3 in the original submission) is generally consistent with that on the full-scale dataset.
> | model | gpt-4o | gpt-4.1 | gpt-4o-mini | Llama-4 | deepseek | Llama-3-8b | Mistral-7B | o3-mini | o4-mini | o3 |
> | --- | --- | --- | --- | --- | --- | --- | --- | --- | --- | --- |
> | Tier 1 | | | | |
> | S-impute | 0.117 | 0.008 | 0.008 | 0.893 | 0.003 | 0.013 | 0.842 | `0.001` | *0.001* | **0.000** |
> | ST-forecast | *0.147* | `0.147` | 0.164 | 0.166 | **0.141** | 0.176 | 0.199 | 0.158 | 0.161 | 0.173 |
> | ST-impute | `0.101` | 0.104 | 0.127 | 0.108 | 0.105 | 0.143 | 0.173 | *0.102* | **0.101** | 0.116 |
> | T-impute | 0.073 | `0.069` | 0.079 | **0.067** | 0.076 | 0.082 | 0.132 | 0.070 | *0.070* | 0.087 |
> | Loc-range | 2.177 | 1.562 | 2.812 | 0.493 | 2.080 | 2.962 | 2.743 | *0.056* | `0.044` | **0.043** |
> | Loc-bearing | 2.895 | 2.620 | 2.926 | 1.688 | 2.223 | 3.032 | 2.813 | *0.166* | **0.151** | `0.153` |
> | Loc-range-bearing | 1.855 | 1.645 | 1.932 | 0.116 | 0.119 | 3.031 | 2.851 | `0.062` | *0.063* | **0.053** |
> | Loc-proximity | 1.065 | 1.034 | 1.054 | 1.034 | 0.958 | 1.413 | 1.801 | *0.956* | `0.878` | **0.858** |
> | T-loc-event | 1.334 | 0.638 | 0.833 | 0.398 | 0.389 | 1.171 | 2.229 | `0.153` | *0.191* | **0.064** |
> | S-loc-event | 2.790 | 2.584 | 2.890 | 2.746 | 2.772 | 2.924 | 2.765 | `0.292` | *0.593* | **0.184** |
> | Track-range | 2.891 | 2.934 | 3.610 | 2.141 | 2.642 | 4.099 | 3.789 | `0.656` | *0.852* | **0.556** |
> | Track-bearing | 3.644 | 3.410 | 3.628 | 2.687 | 2.090 | 4.272 | 3.611 | `0.822` | *0.866* | **0.791** |
> | Track-range-bearing | 1.179 | 2.048 | 1.357 | 0.836 | 0.138 | 4.151 | 3.820 | *0.128* | `0.124` | **0.119** |
> | Track-proximity | **2.020** | `2.024` | 2.097 | 2.209 | 2.156 | 2.657 | 2.763 | 2.276 | 2.205 | *2.083* |
> | S-track-event | 3.340 | 3.750 | 3.619 | 3.493 | 3.647 | 4.040 | 3.488 | `1.265` | *1.815* | **0.931** |
> | T-track-event | 0.329 | *0.136* | 0.255 | 0.447 | 0.141 | 0.383 | 0.428 | `0.114` | 0.138 | **0.080** |
> | Tier 2 | | | | |
> | S relationship | 0.074 | *0.053* | 0.169 | 0.064 | 0.069 | 0.334 | 0.311 | **0.000** | `0.009` | **0.000** |
> | T relationship | **0.000** | **0.000** | `0.021` | **0.000** | **0.000** | 0.467 | *0.363* | **0.000** | **0.000** | **0.000** |
> | ST relationship | 0.133 | *0.042* | 0.187 | 0.214 | 0.130 | 0.434 | 0.507 | **0.026** | `0.029` | 0.057 |
> | Tier 3| | | | |
> | Landmark direction | 0.259 | 0.383 | 0.296 | 0.400 | 0.321 | 0.481 | 0.617 | *0.150* | `0.074` | **0.037** |
> | Intent prediction | `0.207` | 0.332 | 0.398 | 0.253 | 0.237 | 0.506 | 0.361 | 0.282 | *0.224* | **0.195** |
> | Landmark proximity | *0.282* | `0.266` | 0.577 | 0.552 | 0.394 | 0.419 | 0.481 | 0.307 | **0.216** | 0.303 |
> | POI prediction | 0.162 | *0.158* | 0.465 | `0.154` | 0.249 | 0.423 | 0.527 | 0.212 | 0.199 | **0.124** |
> | Route planning | **0.000** | **0.000** | `0.136` | **0.000** | **0.000** | **0.000** | 0.333 | *0.148* | **0.000** | **0.000** |
> | ETA | `0.020` | *0.030* | 0.139 | 0.099 | `0.020` | 0.238 | 0.475 | *0.030* | **0.000** | **0.000** |
> | Route segment dur. | 0.136 | *0.099* | 0.148 | 0.148 | `0.086` | 0.531 | 0.593 | 0.173 | 0.185 | **0.074** |
>
> Furthermore, to support a wider range of research scenarios, **we plan to release both STARK-S (1.3k samples, for low-cost and rapid evaluation) and STARK-L (14k samples, for comprehensive evaluation)**, enabling the community to choose the benchmark size that fits their cost budget.
>
> Lastly, we also provide a detailed breakdown of the evaluation budget per model on STARK-L to guide future users of the benchmark.
>
> | Model         | Cost per run | Provider     |
> |---------------|--------------|--------------|
> | O3*            | $759.8       | OpenAI       |
> | O3-mini       | $442.9       | OpenAI       |
> | O4-mini       | $352.6       | OpenAI       |
> | GPT-4.1       | $237.5       | OpenAI       |
> | GPT-4o        | $284.1       | OpenAI       |
> | GPT-4o-mini   | $18.3        | OpenAI       |
> | LlaMA-4       | $44.4        | Together.ai  |
> | LlaMA-3-8B    | $9.21        | Together.ai  |
> | Mistral-7B    | $25.1        | Together.ai  |
> *OpenAI reduced the cost of o3 by 80% since 06/2025.
>
> >**Q2**: World-knowledge tasks rely on static landmark data, lacking dynamic context (e.g., real-time traffic events).
>
>
> **Response**: We appreciate the reviewer’s observation regarding the lack of dynamic context in world-knowledge tasks. To address this, we introduce a new Incident Detection task designed to evaluate a model’s ability to integrate real-time (dynamic) spatiotemporal context, i.e., traffic incidents.
>
> Specifically, given a route from point A to B, departing at time t, and a set of 20 traffic incidents characterized by {x_i, t_i, e_i} (location, time, and event type),  the model must decide whether any incident affects the route, either **spatially** (occurs along the path) or **temporally** (occurs during the travel window). The key feature of this task is that the **real-time** incident information is injected through the LLM’s context window at inference time, simulating real-world decision-making under evolving conditions. An example is shown as follows:
>
> ```
> Objective:
>
> Let's say I'm travelling from Fisherman's Wharf, San Francisco, CA to Alamo Square (Painted Ladies), San Francisco, CA by car at this start time:9:54 PM, Here is the route I'm taking:
>   0. Start at Location 1
>   1. Go south on Taylor St toward Beach St
>   2. At the traffic light, turn right on Beach St
>   …
>   14. At the traffic light, turn left on Golden Gate Ave
>   15. Finish at Location 2, on the left.
> In addition, there are the following car accidents, represented by location and time pairs:
>   ['29-33 Bellevue Ave, Daly City, California, 94014', '4:41 PM']
>   ['19101-19199 Silver Fox Pl, Castro Valley, California, 94546', '8:15 PM']
> …
>   ['934 S Eldorado St, San Mateo, California, 94402', '7:00 PM']
>   ['Bay Trl, San Rafael, California, 94901', '3:42 PM']
>   ['Bollinger Creek Loop Trl, Lafayette, California, 94549', '4:38 PM']
> Do any of the listed incidents likely impact my route, given the time and locations? You may assume that times for incidents occur on the same day before the current travel start time.
> In addition, you may assume that the car accidents last roughly half an hour before they resolve. Answer directly based on the reasoning of spatial and/or temporal information.
> ```
>
> This task tests the model’s ability to integrate time-varying data provided at query time. We evaluated 11 models on 50 curated instances, and report their error rates below:
>
> | model | gpt-4o | gpt-4.1 | gemini-2.5-flash | gpt-4o-mini | Llama-4 | deepseek-chat | Llama-3-8b | Mistral-7B | o3-mini | o4-mini | o3 |
> | --- | --- | --- | --- | --- | --- | --- | --- | --- | --- | --- | --- |
> | Incident-detection | 0.171 | *0.049* | *0.049* | 0.415 | 0.366 | *0.049* | 0.780 | **0.000** | `0.024` | `0.024` | `0.024` |
>
> Key observations:
> - Models such as O3, O3-mini, GPT-4.1, and Gemini-2.5-flash demonstrate strong reasoning over dynamic spatiotemporal inputs.
> - GPT-4o-mini, LLaMA-4, and especially LLaMA-3-8B underperform, suggesting limited capability in integrating dynamic knowledge.
> - Interestingly, Mistral-7B achieves zero error while LLaMA-3-8B performs poorly, despite similar model sizes. This suggests that architecture or training differences play a significant role; we leave deeper analysis of such small-model divergence for future work.
>
> This task adds a dynamic world-knowledge dimension to the STARK benchmark and reinforces the importance of reasoning over temporally evolving contexts in CPS scenarios.
>
> In the case of the paper being accepted, we will include these new results in the camera-ready version.

---

> > ### Author Response · Authors · 2025-08-04
> > **Follow-up on discussion – Please let us know if any concerns remain**
> >
> > Dear Reviewer,
> >
> > We'd like to kindly follow up on the discussion thread and check if you had any remaining concerns or feedback we could address.
> >
> > As a quick recap:
> >
> > - We completed **full-benchmark evaluations across all models**. The full-scale results are generally consistent with the small-scale evaluation (Table 3, Line 197), validating the statistical robustness of our original findings. To support broader use, we plan to release both STARK-S (1.3k samples) and STARK-L (14k samples), along with a detailed cost breakdown to help researchers manage evaluation scale and budget.
> > - We introduced a new **Incident Detection** task to address your concern regarding static world-knowledge tasks. This task evaluates model reasoning with dynamic, real-time spatiotemporal context injected at inference time.
> >
> > Please let us know if you have any further questions or if there is anything else we can clarify. We look forward to hearing from you.
> >
> > Thanks,
> >
> > Paper 646 authors

---

### Comment · Area_Chair_MGws · 2025-08-09
**Final hours to engage in reviewer-author discussion period**

Dear Reviewers,

please engage with the additional comments supplied by the authors. Note that we are in the final hours of the discussion period. Thank you for your cooperation.

Your AC.

---

### Note · Authors · 2025-08-12

We thank all the reviewers for their valuable comments. In summary, reviewers have many shared positives:
- **All reviewers (7AiH, waoq, oSCC, mhtH) agreed that STARK is novel, well-motivated, and fills a clear gap** in spatiotemporal knowledge reasoning for CPS applications.
- **All reviewers appreciate STARK’s rigorous and hierarchical design**. Besides, STARK is the first to combine ArcGIS DE-9IM + Allen algebra and systematically evaluate direct answering (DA) and code interpreter (CI) modes.
- **All reviewers recognize the comprehensive evaluation and insights**. The results of 3 LRMs and 8 LLMs identified the strengths and limits of both architectures, and the results of DA and CI reveal insights into tool-aided LLM agents.

We also received several constructive comments. Below are the key questions and our responses:
- **Partial benchmark evaluation due to cost constraints (7AiH, waoq, mhtH)**. Following the cost reduction of o3, we conducted full-benchmark evaluations on all models (except GPT-4.5 being deprecated). Results confirm consistency between small- and full-scale evaluations. We will release both STARK-S (1.3k, rapid use) and STARK-L (14k, complete evaluation) for flexible use. Though reviewer 7AiH did not follow up, other reviewers acknowledged the resolution, with waoq raising the score and mhtH marking the response satisfactory.
- **To address the concern about relying on closed-source LRMs (waoq, mhtH), we added open-source LRM experiments**. The insights in the submission regarding the positive correlation between model size and performance are further supported, with both reviewers marking the response satisfactory.
- **World-knowledge reasoning relies only on static landmark data, lacking dynamic context (7AiH)**. We added a new Incident Detection task, introducing real-time traffic events at inference time, testing models’ ability to integrate dynamic world knowledge.
- **No systematic CI failure analysis (waoq)**. We introduced a taxonomy of 6 root and side failures, offering practical recommendations for improving the spatiotemporal reasoning abilities of tool-aided agents.

All reviewers who responded were satisfied with our rebuttals. Reviewer 7AiH did not engage after the initial review, but we believe we addressed their concerns by full benchmark evaluation and a new Incident Detection task. Here, we sincerely thank all reviewers and will include all new experiments, analysis, and insights if the paper is accepted.

---

### Decision · Program_Chairs · 2025-09-18

**Decision:**

Accept (poster)

**Comment:**

The paper makes a substantial contribution by introducing STARK, a pioneering benchmark that systematically evaluates spatiotemporal reasoning capabilities of LLMs and LRMs. Its hierarchical task structure—state estimation, temporal reasoning, and knowledge-enhanced reasoning—captures the full perception-to-decision pipeline, offering more depth and realism than prior benchmarks like GeoQA. The benchmark is impressively comprehensive, with 26 tasks spanning 14,552 instances across five sensor modalities (range, bearing, event, proximity, status), and includes realistic noise to reflect practical conditions. Methodologically, STARK emphasizes open-ended answers and tool-use evaluation, making it more representative of real-world CPS applications. The experiments are thorough, with detailed comparisons showing LRMs’ robustness and highlighting scaling trends in LLMs (e.g., GPT-4.5 with code interpreter). Importantly, the authors release the dataset, code, evaluation scripts, and API usage costs, making the work a transparent and valuable open-source resource for future research.
The weaknesses of the paper that were raised by the reviewers were mainly cosmetics / presentations and clarifications, and those were addressed during the rebuttal and discussion period. Apart from reviewer 7AiH who was non-responsive, all other reviewers vouched for acceptance.

===== FINAL UPDATE FROM DB Track PCs ====

The final decision for this paper has been taken by the program chairs after consultation with the SACs. All Senior Area Chairs have ranked papers according to the feedback from the AC during the review process. We decided to leave the original meta-review to reflect the opinion of the AC in light of the initial discussions with reviewers and SAC.